# AffineQuant: Affine Transformation Quantization for Large Language Models

**Yuexiao Ma**[1][*] **Huixia Li**[2], **Xiawu Zheng**[1,3,4], **Feng ling**[2], **Xuefeng Xiao**[2],
**Rui Wang**[2], **Shilei Wen**[2], **Fei Chao**[1], **Rongrong Ji**[1,4][†]
[1] Key Laboratory of Multimedia Trusted Perception and Efficient Computing,
Ministry of Education of China, School of Informatics, Xiamen University, 361005, P.R. China.
[2] ByteDance Inc. [3] Peng Cheng Laboratory, Shenzhen, China.
[4] Institute of Artificial Intelligence, Xiamen University.

## Abstract

The significant resource requirements associated with Large-scale Language Models (LLMs) have generated considerable interest in the development of techniques aimed at compressing and accelerating neural networks. Among these techniques, Post-Training Quantization (PTQ) has emerged as a subject of considerable interest due to its noteworthy compression efficiency and cost-effectiveness in the context of training. Existing PTQ methods for LLMs limit the optimization scope to scaling transformations between pre- and post-quantization weights. This constraint results in significant errors after quantization, particularly in low-bit configurations. In this paper, we advocate for the direct optimization using equivalent Affine transformations in PTQ (AffineQuant). This approach extends the optimization scope and thus significantly minimizing quantization errors. Additionally, by employing the corresponding inverse matrix, we can ensure equivalence between the pre- and post-quantization outputs of PTQ, thereby maintaining its efficiency and generalization capabilities. To ensure the invertibility of the transformation during optimization, we further introduce a gradual mask optimization method. This method initially focuses on optimizing the diagonal elements and gradually extends to the other elements. Such an approach aligns with the Levy-Desplanques theorem, theoretically ensuring invertibility of the transformation. As a result, significant performance improvements are evident across different LLMs on diverse datasets. Notably, these improvements are most pronounced when using very low-bit quantization, enabling the deployment of large models on edge devices. To illustrate, we attain a C4 perplexity of 15.76 (2.26↓ vs 18.02 in OmniQuant) on the LLaMA2-7B model of W4A4 quantization without overhead. On zero-shot tasks, AffineQuant achieves an average of 58.61% accuracy (1.98% ↑ vs 56.63 in OmniQuant) when using 4/4-bit quantization for LLaMA-30B, which setting a new state-of-the-art benchmark for PTQ in LLMs.

## 1 Introduction

Large Language Models (LLMs) (Zhang et al., 2022; Touvron et al., 2023a;b) attract increasing attention due to their impressive performance. However, emergent logical reasoning abilities (Wei et al., 2022a) are only present in models above a certain size threshold. Hence, the training and inference efficiency of LLMs necessitates careful consideration. Specifically, the potential utilization of LLMs for inference on mobile and edge devices drives our motivation to focus on accelerating model inference. Quantization is regarded as one of the most promising methods among these compression methods. In particular, it maps weights or activations to lower bit representations, effectively reducing the memory usage of the model. Additionally, optimizing the compilation of operators for low-bit operations (MLC, 2023) significantly enhance their efficiency and accelerate model inference.

---

[*]This work was done when Yuexiao Ma was intern at ByteDance Inc.
Code is available at: `https://github.com/bytedance/AffineQuant`
[†]Corresponding Author: rrji@xmu.edu.cn

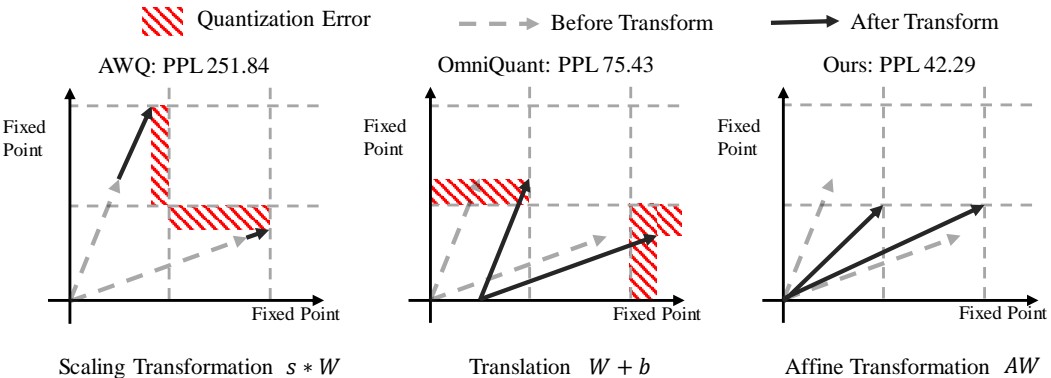

Figure 1: The effect of scaling, translation and affine transformation on the quantization of the weights. The term "Fixed Point" refers to the $2^n - 1$ quantization levels in $n$-bit quantization. $s$, $b$, and $A$ are the scaling factor, translation factor, and affine transformation matrix, respectively. We assume that the input channel and output channel of $W$ is 2. We consider each output channel as a two-dimensional vector.

Meanwhile, due to the substantial computational resources and high-quality data required for training LLMs (Zhang et al., 2023a;c), implementing quantization through model fine-tuning is challenging. Therefore, the research community is increasingly emphasizing training-free algorithms, termed as Post-Training Quantization (PTQ) (Wei et al., 2022c; Xiao et al., 2023; Wei et al., 2023; Lin et al., 2023; Shao et al., 2023; Yuan et al., 2023). Post-training quantization allows for efficient optimization with less calibration data. However, this process can lead to significant performance degradation, particularly in small-sized models or low-bit scenarios.

Equivalent transformations are widely adopted in most PTQ methods. As illustrated in Figure 1, AWQ (Lin et al., 2023) enhances scale computation by optimizing statistics and introduces the mean square error loss between the pre- and post-quantization feature maps as an optimization metric for the first time in LLMs. Recently, Omniquant (Shao et al., 2023) introduces block-wise learnable scale and shift parameters for enhanced optimization. In higher dimensions, the concept of equivalent quantization is also gaining attention. RPTQ (Yuan et al., 2023) achieves activation quantization per-cluster by sorting the columns of activation values. The reordering can be mathematically represented by converting the scale from a vector into a matrix form, where each row and column corresponds to a single scale value. This transformation effectively rearranges the activation columns and weight rows in an equivalent manner.

In summary, the evolution of equivalent quantization progresses from manual design to gradient optimization, from low-dimensional to high-dimensional transformations, and from single-scale merging to a combination of multiple operations, including translation and reordering. Equivalent quantization offers advantages in two main aspects. Firstly, by ensuring consistency between the pre- and post-quantization outputs, the introduced quantization noise can be effectively mitigated through optimization of the equivalence transform parameters. This aligns with the concept of post-training quantization, where the equivalence transform acts as an intermediate agent for noise improvement. Secondly, different types of equivalence transforms are orthogonal to each other. Intuitively, the introduction of each new type of equivalence transform expands the parameter optimization space, resulting in performance improvements.

Therefore, we propose an algorithm for equivalent affine transformation. Specifically, we left-multiply the affine transform matrix to weights in the linear layer and right-multiply the activations with the transform matrix inverse. Guided by the mean square error loss, we optimize the affine transformation matrix, resulting in consistently lower loss compared to other algorithms on a wide range of models during the optimization process. Furthermore, we explore the invertibility of matrices during the optimization process. The Levy-Desplanques theorem (Naimark & Zeheb, 1997) demonstrates that the strictly diagonally dominant matrices are invertible. To ensure that the affine transformation matrix is strictly diagonally dominant, we employ diagonal initialization and gradual mask methods. In this way, the optimization of high-dimensional matrices with limited calibra-

tion data is stabilized through a gradual optimization process that involves freezing the parameters. In terms of inference efficiency, our method is consistent with other methods after matrix merging. Finally, our method achieves state-of-the-art performance in LLMs quantization, particularly in scenarios involving small-scale models or lower bit configurations. Overall, our contributions are summarized as follows:

- We propose a novel affine transform in PTQ, which retains the benefits of PTQ, assures efficiency and generalization, significantly minimizes quantization error, especially under low-bit quantization, and enables the deployment of LLMs on edge devices.

- We propose a novel optimization algorithm that guarantees invertibility throughout the process, utilizing the Levy-Desplanques theorem, and simultaneously reduces computational costs.

- Our method obtains the state-of-the-art performance for large language model quantization, especially on low-bit or small models. Without additional overhead, on the w4a4 configuration of LLaMA2-7B, our perplexity on the C4 dataset is 15.76 (2.26↓ vs 18.02 in OmniQuant). Similarly, on the w4a4 configuration of LLaMA-30B, our accuracy on 6 zero-shot tasks is 58.61% (1.98 ↑ vs 56.63 in OmniQuant).

## 2 RELATED WORK

Quantization can be classified into two main categories based on algorithmic efficiency and data requirements: Quantization-Aware Training (QAT) and Post-Training Quantization (PTQ). QAT (Bondarenko et al., 2023; Choi et al., 2018; Yao et al., 2021; Gong et al., 2019; Wang et al., 2019; Esser et al., 2019; Sun et al., 2020; Lee et al., 2021) requires a substantial amount of data for fine-tuning the model weights, making it challenging to support LLMs (Zhang et al., 2023b). In contrast, we focus on PTQ (Nagel et al., 2020; Li et al., 2021; Wei et al., 2022b; Ma et al., 2023; Liu et al., 2021; Cai et al., 2020; Cheng et al., 2023) in this study due to its efficient algorithmic approach.

**Post-Training Quantization (PTQ).** In the field of CNN, PTQ primarily focuses on optimizing weight rounding strategies. Adaround (Nagel et al., 2020) improves quantization models through optimized weight rounding, considering rounding up or down. BRECQ (Li et al., 2021) introduces a block-wise optimization process and incorporates squared gradient information. QDROP (Wei et al., 2022b) enhances the performance of quantized models by randomly drop quantized activations.

**Large Language Model Quantization.** To address computational resource constraints, the community has focused on efficient quantization algorithms for Large Language Models (LLMs). LLMs quantization can be categorized into weight-only quantization (Frantar & Alistarh, 2022; Lin et al., 2023; Shao et al., 2023; Kim et al., 2023) and weight-activation quantization (Xiao et al., 2023; Wei et al., 2023; Yao et al., 2022; Dettmers et al., 2022; Shao et al., 2023), depending on whether activations are quantized. Given the large size of LLMs, memory access efficiency becomes a primary bottleneck for acceleration. Weight-only quantization addresses this by compressing model weights to lower bit precision, effectively mitigating the memory wall problem (Kim et al., 2023).

## 3 METHODOLOGY

In this section, we introduce AffineQuant, an approach that utilizes equivalent affine transformation for quantization. Compared to other methods, AffineQuant consistently maintains optimal mean square error throughout the optimization process. We also explore the reversibility of the affine transform matrix during optimization. To ensure stability, we propose a gradual masking approach based on the Levy–Desplanques theorem (Naimark & Zeheb, 1997) to maintain the affine transform matrix as a strictly diagonally dominant matrix. Lastly, we analyze the inference efficiency of LLMs following the optimization performed by AffineQuant.

### 3.1 AFFINEQUANT

When considering the concept of equivalent transformations from a physical perspective, we can draw analogies to certain operations. For instance, in SmoothQuant (Xiao et al., 2023), we can analogize scale to scaling operations for vectors, while in Outlier Suppression+ (Wei et al., 2023),

we can analogize shift to translation operations for vectors. Similarly, rotations of vectors can also be classified as equivalent transformations.

We define the pseudo-quantization function as follows:

$$\mathcal{Q}(x) = \Delta * \left( clamp \left( \left\lfloor \frac{x}{\Delta} \right\rceil + zp, 0, 2^n - 1 \right) - zp \right),$$

(1)

where $\Delta$, $zp$ and $n$ are the quantization step-size, zero point and bits, respectively. $\lfloor \cdot \rceil$ is the rounding operation. As depicted in Figure 1, AffineQuant involves left-multiplying the affine transform matrix $A$ by weight matrix $W$ to better align the weight distribution with the quantization function $Q(\cdot)$. Expanding the optimization space enables smaller quantization errors in the transformed weights, leading to a reduction in perplexity. Simultaneously, we right-multiply the inverse of the affine transform matrix $A$ by the activation value $X$ to maintain the invariance of the matrix multiplication output between activations and weights. For a single linear layer, AffineQuant formulates the following optimization problem:

$$\arg\min_{A} \left\| XW - XA^{-1}\mathcal{Q}(AW) \right\|_F^2.$$

(2)

AffineQuant incorporates the essence of AWQ (Lin et al., 2023) and SmoothQuant (Xiao et al., 2023) when the main diagonal elements of the matrix $A$ are computed from weight and activation statistics. It aligns with OmniQuant (Shao et al., 2023) by exclusively updating the diagonal elements of $A$. The reordering matrices used in RPTQ (Yuan et al., 2023) are a subset of the affine transformation matrix $A$ when each row and column of $A$ contains only one occurrence of the element 1. In summary, AffineQuant encompasses various previous equivalent quantization algorithms, thereby expanding the optimization possibilities for the weight distribution $W$.

In Figure 1, let the weight matrix $W \in \mathbb{R}^{2\times2}$ have 2 output channels and input channels. The scaling factor, translation factor, and affine transformation matrix are denoted as $s$, $b$, and $A$, respectively. We divide the weight matrix into 2 vectors $\{v_1, v_2\}$ based on output channels. The scaling transform $s_i * v_i$ uniformly scales each element of $v_i$. The translation transform $v_i + b_i$ shifts $v_i$ along different axes. The affine transformation $Av_i$ allows for arbitrary repositioning of $v_i$. However, the scaling and translation transformations are limited in their ability to map dimensions in $v_i$ to adjacent quantized fixed points. In contrast, the affine transformation guarantees convergence of all dimensions in a vector to the quantized fixed point. In other words, the affine transformation aligns the weight distribution with the noise introduced by the quantization function $\mathcal{Q}(x)$ in Equation 2, resulting in reduced quantization error. It is worth noting that normalizing the affine transformation matrix by rows $\left( A \to s^{'} A^{'} \right)$, where each row of the matrix $A^{'}$ has a norm of 1, transforms $A^{'}$ into a standard rotation matrix. This rotation matrix rotates the output channels of the weights while preserving their magnitudes. The scaling factor $s^{'}$ performs scaling on the rotated vectors. Therefore, the affine transformation matrix $A$ combines both scaling and rotation equivalent transformations and is orthogonal to the translation transformation.

The perplexity (ppl) exhibits an exponential relationship with the cross-entropy (CE) loss, which is positively correlated with the mean square error of the output activation before and after quantization, as demonstrated in (Nagel et al., 2020; Li et al., 2021). Hence, optimizing perplexity can be achieved by optimizing the mean square error before and after quantization. Specifically,

$$PPL \propto \mathcal{L}_{CE} \propto \left\| XW - XA^{-1}\mathcal{Q}(AW) \right\|_F^2,$$

(3)

In large language models quantization, the optimization objective of AffineQuant is as follows,

$$\arg\min_{A,\delta} \left\| f_i\left(X, W\right) - f_i\left(\left(X - \delta\right) A^{-1}, \mathcal{Q}\left(AW\right), b + \delta W\right) \right\|_F^2.$$

(4)

where $f_i$ is the $i$-th transformer block. $(X - \delta) A^{-1}$, $\mathcal{Q}(AW)$, $b + \delta W$ are the activation, weight and bias after the equivalent transformation, respectively. We combine the affine and translation transformations and use the mean square error of the transformer block output, both pre- and post-quantization, as the optimization objective.

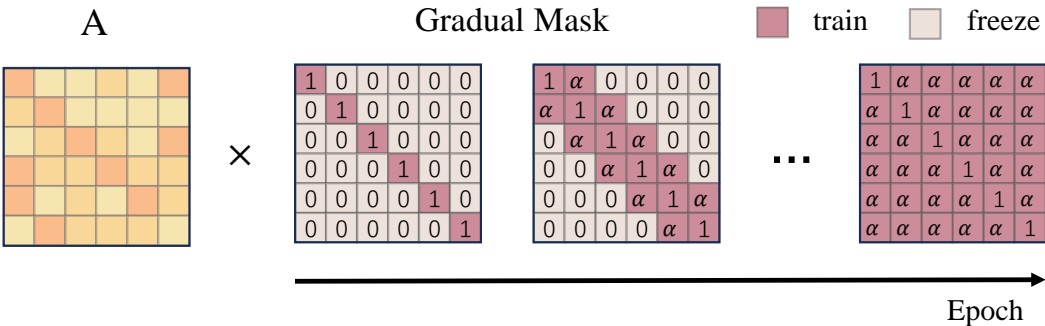

Figure 2: The gradual mask operates on the affine transformation matrix, gradually incorporating the elements of matrix $A$ near the diagonal into the training process as training progresses.

Figure 3 illustrates the mean square error loss optimization for the last transformer block of LLaMA-7B and OPT-1.3B. Notably, AffineQuant exhibits a lower initial loss compared to OmniQuant (Shao et al., 2023) due to the superior performance of the affine transformation matrices in the preceding blocks. Additionally, AffineQuant demonstrates faster loss convergence and superior overall optimization performance in the last block compared to OmniQuant. These results reaffirm the significant potential of invertible matrix optimization. Figure 5 and 6 in Appendix A.4 presents a random sampling of multiple stability factors ($alpha$), which impact on quantization loss convergence. The data reveals a notable link between the last transformer block's quantization loss and the quantized model's performance. This implies AffineQuant's effectiveness in reducing quantization loss in Figure 3, thereby enhancing the model's quantization performance during optimization.

## 3.2 Reversibility and Gradual Mask

In the optimization process, it is necessary to invert the affine transformation matrix. However, we do not include any constraints in the objective function (Equation 4) to ensure the matrix remains full rank or well-conditioned. Therefore, how to keep the matrix invertible during the optimization process? To begin, let's define a strictly diagonally dominant matrix as follows:

**Definition 1** *(Strictly Diagonally Dominant Matrix) A matrix $A$ is considered strictly diagonally dominant if the absolute value of each diagonal element is greater than the sum of the absolute values of the remaining elements in the corresponding row. Specifically,*

$$|a_{ii}| > \sum_{i \neq j} |a_{ij}|, \quad for\ all\ i. \tag{5}$$

The Levy-Desplanques theorem (Naimark & Zeheb, 1997) establishes that all strictly diagonally dominant matrices are invertible. By initializing the affine transformation matrix with diagonal elements, we ensure it initially is strictly diagonally dominant. Although utilizing second-order momentum and lower learning rates in the optimizer can assist in satisfying the requirements of the Levy-Desplanques theorem, the optimization of large affine transform matrices still faces instability challenges as the model size increases.

To ensure that the affine transformation matrix remains strictly diagonally dominant during optimization, we introduce a gradual mask approach, as illustrated in Figure 2. At the start of each optimization block, we freeze all elements except for those on the main diagonal. As the optimization progresses, we gradually unfreeze the elements near the main diagonal. Eventually, all matrix elements become learnable for optimization. This freezing mechanism, referred to as the **G**radual **M**ask (GM), is defined as follows:

$$GM_{ij} = \begin{cases} 1 & i = j, \\ \alpha & 0 < |i - j| \leq \frac{e}{t} \times hidden\ size, \\ 0 & otherwise, \end{cases} \tag{6}$$

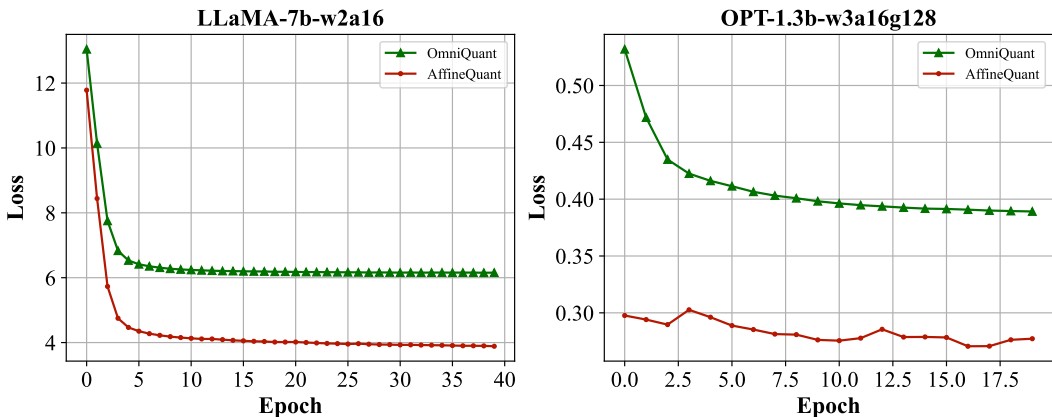

Figure 3: Mean square error loss of the last transformer block of LLaMA-7b and OPT-1.3b. "w2a16" means 2-bit weight-only quantization. "w3a16g128" means 3-bit grouping 128 weight-only quantization. We optimize 40 and 20 epochs in the last block of LLaMA-7b and OPT-1.3b, respectively.

Where $GM_{ij}$ is the $i$-th row, $j$-th column element of the mask matrix. $t$ is the target epochs. $e \in [1, t]$ is the current epochs. "hidden size" is the dimension of the affine transformation matrix. $\alpha$ is the stability factor. Within the attention module, we apply a gradual mask in each attention head. GM is a learning rate regulator that achieves its purpose by element-wise dot-producting with the matrix $A$. Specifically, the impact of the GM matrix on the optimization process can be divided into two aspects. Here, we present the optimization process for the matrix A after incorporating the GM.

$$\textbf{Forward:} \quad A_e^* = A_e \circ GM_e, \tag{7}$$

$$\textbf{Backward:} \quad A_{e+1} = A_e + \eta \frac{\partial L}{\partial A_e^*} \frac{\partial A_e^*}{\partial A_e}, \tag{8}$$

$$= A_e + \eta GM_e \frac{\partial L}{\partial A_e^*}. \tag{9}$$

Where $\circ$ is the Hadamard product. $A_e$ and $GM_e$ are the matrices $A$ and Gradual Mask (GM) matrix in epoch $e$, respectively. $\eta$ is the learning rate of matrix $A$. $L$ is the optimization loss. The GM matrix effectively reduces the magnitude of non-principal diagonal elements in matrix $A$ during forward propagation when the stability factor $\alpha$ is less than 1. This ensures the existence of a stable inverse matrix of $A^*$ in the optimization process during epoch $e$, as per the Levy-Desplanques theorem. In backward propagation, GM affects the learning rate $\eta$, thereby suppressing the update rate of non-primary diagonal elements in matrix $A$. Consequently, the impact of GM on $\eta$ ensures that matrix $A$ in epoch $e + 1$ maintains strictly diagonally dominant, satisfying the Levy-Desplanques theorem. Notably, as $\alpha$ approaches 0, the optimization process converges stably and becomes equivalent to OmniQuant (Shao et al., 2023). Additionally, Appendix A.2 includes a theorem demonstrating that a sufficiently small stability factor $\alpha$ ensures the strictly diagonal dominance of matrix A during optimization.

The concept of gradual adaptation is also present in the post-training quantization of vision models. In Adaround (Nagel et al., 2020), the gradient update of parameters is controlled using gradual powers of $\beta$ in the soft quantization function. When $\beta$ is sufficiently large, only values close to 0 or 1 are updated due to gradient limitations. As optimization progresses, the gradient of all rounded values is gradually released. However, it is important to note that the goals and approaches of the two methods are distinct. Adaround (Nagel et al., 2020) employs the gradual power $\beta$ to prevent fast convergence of the objective function, which can lead to suboptimal optimization. On the other hand, the gradual mask in AffineQuant ensures the strictly diagonally dominant property of the affine transformation matrix. Appendix A.6 showcases heat maps depicting different block affine transformation matrices at various epochs, demonstrating the effectiveness of the gradual mask approach in maintaining strictly diagonally dominant matrices.

Table 1: Weight-only quantization PPL(↓) results on the OPT model WikiText2 dataset.

| Config | Method | 125M | 1.3B | 2.7B | 6.7B | 13B | 30B |
|--------|--------|------|------|------|------|-----|-----|
| FP16 | - | 27.65 | 14.63 | 12.47 | 10.86 | 10.12 | 9.56 |
| w3a16 | RTN | 1.2e3 | 1.3e4 | 1.6e4 | 6.5e3 | 4.6e3 | 1.5e3 |
| | GPTQ (Frantar et al., 2022) | 53.05 | 21.17 | 16.83 | 15.09 | 11.73 | 10.30 |
| | AWQ (Lin et al., 2023) | 69.43 | 28.01 | 263.10 | 15.13 | 20.09 | 35.74 |
| | OmniQuant (Shao et al., 2023) | 35.66 | 16.68 | 13.80 | 11.65 | 10.87 | 10.00 |
| | AffineQuant | 30.56 | 15.94 | 13.15 | 11.44 | 10.76 | 9.98 |
| w3a16g128 | RTN | 51.22 | 119.00 | 297.98 | 23.54 | 46.03 | 18.80 |
| | GPTQ (Frantar et al., 2022) | 39.24 | 16.47 | 13.69 | 11.65 | 10.35 | 9.73 |
| | AWQ (Lin et al., 2023) | 36.74 | 16.32 | 13.58 | 11.41 | 10.68 | 9.85 |
| | OmniQuant (Shao et al., 2023) | 32.25 | 15.72 | 13.18 | 11.27 | 10.47 | 9.79 |
| | AffineQuant | 30.21 | 15.61 | 12.98 | 11.18 | 10.51 | 9.81 |
| w4a16 | RTN | 37.28 | 48.17 | 16.92 | 12.10 | 11.32 | 10.97 |
| | GPTQ (Frantar et al., 2022) | 31.43 | 15.56 | 12.82 | 11.41 | 10.31 | 9.63 |
| | AWQ (Lin et al., 2023) | 32.28 | 15.49 | 12.93 | 11.30 | 10.39 | 9.77 |
| | OmniQuant (Shao et al., 2023) | 29.45 | 15.04 | 12.76 | 11.03 | 10.30 | 9.65 |
| | AffineQuant | 28.39 | 14.92 | 12.64 | 10.96 | 10.26 | 9.65 |
| w4a16g128 | RTN | 30.47 | 15.29 | 13.02 | 11.15 | 10.30 | 9.94 |
| | GPTQ (Frantar et al., 2022) | 29.81 | 14.89 | 12.52 | 10.93 | 10.17 | 9.58 |
| | AWQ (Lin et al., 2023) | 29.15 | 14.94 | 12.74 | 10.93 | 10.21 | 9.59 |
| | OmniQuant (Shao et al., 2023) | 28.86 | 14.88 | 12.65 | 10.96 | 10.20 | 9.62 |
| | AffineQuant | 28.33 | 14.79 | 12.58 | 10.92 | 10.19 | 9.62 |

Table 2: AffineQuant and OmniQuant quantization performance of LLaMA-7B, 13B, 30B on six zero-shot datasets using 4/4 bit quantization.

| | Dataset | PIQA (↑) | ARC-e (↑) | WinoGrande (↑) | BoolQ (↑) | ARC-c (↑) | HellaSwag (↑) | Avg. (↑) |
|--|---------|------|-------|------------|-------|-------|-----------|------|
| LLaMA-7B w4a4 | FP16 | 77.47 | 52.48 | 67.07 | 73.08 | 41.46 | 73.00 | 64.09 |
| | OmniQuant Shao et al. (2023) | 66.15 | 45.20 | 53.43 | 63.51 | 31.14 | 56.44 | 52.65 |
| | AffineQuant | 69.37 | 42.55 | 55.33 | 63.73 | 31.91 | 57.65 | 53.42 |
| LLaMA-13B w4a4 | FP16 | 79.10 | 59.89 | 70.31 | 68.01 | 44.45 | 76.21 | 66.33 |
| | OmniQuant Shao et al. (2023) | 69.69 | 47.39 | 55.80 | 62.84 | 33.10 | 58.96 | 54.37 |
| | AffineQuant | 66.32 | 43.90 | 54.70 | 64.10 | 29.61 | 56.88 | 52.58 |
| LLaMA-30B w4a4 | FP16 | 80.08 | 58.92 | 72.53 | 68.44 | 45.47 | 79.21 | 67.44 |
| | OmniQuant Shao et al. (2023) | 71.21 | 49.45 | 59.19 | 65.33 | 34.47 | 64.65 | 56.63 |
| | AffineQuant | 70.84 | 49.41 | 58.64 | 70.12 | 37.12 | 65.53 | 58.61 |

## 3.3 EFFICIENCY

**Optimize Efficiency.** PyTorch's linear algebra library (Paszke et al., 2019) offers matrix inverse computations in both float and double precision. Consequently, we maintain the model's precision as either float or double throughout the optimization process. Furthermore, approximate computations of the matrix inverse may contain errors due to the numerical precision limitations of the computer. Therefore, we analyze memory consumption, optimization time, error magnitude, and the impact on model performance for both precision types in Section 4.3.

**Inference Efficiency.** In line with similar algorithms, we integrate the affine transformation matrix with other layers. Subsequently, we perform half-precision inference on the network. For all linear layers, we merge the affine transformation matrix with the weight and bias parameters. In addition, we only optimize the diagonal elements of the affine matrix after LayerNorm for weight-activation quantization. This allows us to merge the affine matrix with the LayerNorm weights and bias. Consequently, AffineQuant can be achieved without introducing any additional overhead to model inference. Tables 2 and 3 demonstrate AffineQuant's superior performance over other methods in zero-shot and PPL tasks, even without additional overhead, using 4/4-bit quantization.

Table 3: Quantization performance of LLaMA1&2 on WikiText2 and C4 datasets using 4/4 bit weight-activation quantization.

| Datasets | LLaMA1&2 | Methods | 1-7B | 1-13B | 1-30B | 2-7B | 2-13B |
|---|---|---|---|---|---|---|---|
| WikiText2 | FP16 | - | 5.68 | 5.09 | 4.10 | 5.47 | 4.88 |
| | W4A4 | SmoothQuant Xiao et al. (2023) | 25.25 | 40.05 | 192.40 | 83.12 | 35.88 |
| | | OmniQuant Shao et al. (2023) | 11.26 | 10.87 | 10.33 | 14.26 | 12.30 |
| | | AffineQuant | 10.28 | 10.32 | 9.35 | 12.69 | 11.45 |
| C4 | FP16 | - | 7.08 | 6.61 | 5.98 | 6.97 | 6.46 |
| | W4A4 | SmoothQuant Xiao et al. (2023) | 32.32 | 47.18 | 122.38 | 77.27 | 43.19 |
| | | OmniQuant Shao et al. (2023) | 14.51 | 13.78 | 12.49 | 18.02 | 14.55 |
| | | AffineQuant | 13.64 | 13.44 | 11.58 | 15.76 | 13.97 |

Table 4: PPL, memory usage, optimization runtime, and merge error for the OPT model under three precision schemes. The "double" scheme maintains double precision for both the model and the transform matrix. The "float" scheme indicates that both the model and the transform matrix are in float precision. The "float-double" scheme denotes that the model is in float precision while the transform matrix is in double precision.

| | Merge Error | OPT-125M w2a16g64 | | | OPT-6.7B w4a16 | | |
|---|---|---|---|---|---|---|---|
| | | PPL | Memory Utilization | Runtime | PPL | Memory Utilization | Runtime |
| FP16 | - | 24.60 | - | - | 11.74 | - | - |
| Double | 1.88e−16 | 42.43 | 7065.5Mb | 1.19h | 11.91 | 41414.3Mb | 16.7h |
| Float | 2.58e−3 | 42.91 | 3586.6Mb | 0.78h | 11.90 | 21188.9Mb | 8.65h |
| Float-Double | 3.48e−4 | 42.88 | 3663.6Mb | 0.85h | 11.96 | 23189.5Mb | 12.72h |

# 4 EXPERIMENTS

## 4.1 IMPLEMENTATION DETAILS

**Algorithm Details.** In AffineQuant, the stability factor $\alpha$ decreases as the model size increases, the quantization bits decrease, and the group size increases. For OPT-6.7B and smaller models, we set $\alpha = 1$. As the model size increases, we use $\alpha = 1e - 2$ for configurations with weight quantization of 3 bits or more. For other configurations, we select $\alpha$ from the set $\{1e - 2, 1e - 3, 1e - 4\}$. Then, we exclude the affine transformation between the two linear layers in the MLP module. This is because the optimization of the large transformation matrix in inflated dimensions is challenging. Additionally, the presence of the activation function renders the equivalent transformation of the matrix invalid.

## 4.2 EVALUATION EXPERIMENTS

As demonstrated in Tables 1 and 3, we observe consistent performance improvements across all models with various quantization configurations. This indicates that AffineQuant is not reliant on a particular quantization configuration. Notably, AffineQuant exhibits significant improvements, particularly in cases of low-bit quantization or smaller model sizes. Specifically, in the w3a16g128 configuration on the OPT-125M model, we achieve a perplexity reduction of 5.10, surpassing the performance of OmniQuant (Shao et al., 2023) by a large margin. Furthermore, we achieve perplexity reductions of 2.26 and 1.57 on LLaMA2-7B models with C4 and WikiText2 datasets, under the w4a4 quantization configuration. The aforementioned results underscore the importance of expanding the optimization space for the equivalence factor in challenging quantization tasks. For additional dataset results, please refer to the Appendix.

## 4.3 ABLATION STUDY

**Impact of numerical precision.** We compare various metrics, including merge error, PPL, memory usage, and optimization runtime, for different precision schemes. Specifically, in the "float-double" scheme, we convert the activations or weights to double precision, multiply them with the trans-

Table 5: Effect of different stability factors $\alpha$ on model performance of OPT-125M and LLaMA-7B.

|  | Dataset | FP16 | 1e0 | 1e−1 | 1e−2 | 1e−3 | 1e−4 | 1e−5 | 1e−6 | 1e−7 | 1e−8 |
|---|---|---|---|---|---|---|---|---|---|---|---|
| OPT-125M w2a16g128 | WikiText2 | 27.65 | 42.08 | 45.32 | 65.77 | 76.03 | 76.78 | 76.00 | 76.48 | 76.55 | 76.46 |
|  | PTB | 32.54 | 63.60 | 68.29 | 114.41 | 135.72 | 125.38 | 124.12 | 132.36 | 132.45 | 113.75 |
|  | C4 | 24.60 | 45.80 | 50.84 | 70.44 | 79.60 | 81.48 | 79.72 | 79.67 | 80.70 | 79.54 |
| LLaMA-7B w2a16 | WikiText2 | 5.68 | NaN | NaN | 9.53 | 10.63 | 10.90 | 10.99 | 10.72 | 11.63 | 11.67 |
|  | C4 | 7.08 | NaN | NaN | 14.89 | 14.62 | 15.31 | 15.22 | 14.78 | 16.66 | 17.33 |

Table 6: Contributions of gradual mask on OPT-125M and LLaMA-7B model.

|  | Scheme | WikiText2 | PTB | C4 |  | Scheme | WikiText2 | C4 |
|---|---|---|---|---|---|---|---|---|
| OPT-125M w3a16 | FP16 | 27.65 | 32.54 | 24.60 | LLaMA-7B w2a16 | FP16 | 5.68 | 7.08 |
|  | With Gradual | 32.10 | 39.85 | 29.97 |  | With Gradual | 9.53 | 14.89 |
|  | Without Gradual | 53.52 | 90.47 | 62.17 |  | Without Gradual | NaN | NaN |

formation matrix, and then truncate them to float precision. For the merge error, we define two linear layers with input and output channels set to $4,096$. We randomly sample the affine transformation matrix $A \in \mathbb{R}^{4096 \times 4096}$ and the input activation $X \in \mathbb{R}^{2048 \times 4096}$. We conduct $1,000$ runs to calculate the mean square error averages of the linear layer outputs before and after merging $A$ for different precision schemes. The results are presented in Table 4. Notably, in the case of double precision optimization, the computational error of the matrix inverse is minimized. Despite the higher time and memory usage compared to other schemes, the smaller merge error results in a minor improvement in PPL. However, this improvement is not significant for larger-scale models.

**Effects of stability factor.** In Table 5, we adjust the stability factor $\alpha$ in Equation 6. The affine transform theoretically converges to the scale transform as $\alpha$ approaches 0. We observe that as $\alpha$ decreases, the model performance of OPT-125M and LLaMA-7B converges to OmniQuant (Shao et al., 2023). However, in the case of LLaMA-7B, a larger stability factor does not guarantee the strictly diagonal dominance of the transformation matrix, which can lead to training collapse. Hence, it is necessary to increase $\alpha$ while ensuring the validity of the Levy-Desplanques theorem (Naimark & Zeheb, 1997).

**Contribution of gradual mask.** In Equation 6, we gradually release the elements of the mask matrix close to the diagonal. In Table 6, when we remove the gradual approach, the OPT-125M and LLaMA-7B models exhibit poor performance or fail to complete training. This indicates that updating all parameters of the affine transformation matrix at the outset is not conducive to maintaining its invertible or well-conditioned properties. The gradual mask approach provides a stable means to optimize large matrices.

## 5 CONCLUSION

Post-training quantization based on equivalence shows significant potential. However, previous equivalence methods have limited the optimizable weight space, resulting in a large quantization error in the transformed weight distribution. This limitation becomes more pronounced in the case of small models and low-bit quantization. Affine transformation methods address this issue by significantly expanding the optimizable weight space. Previous transformation methods can be seen as a special case of affine transformation or orthogonal to it. Moreover, the Levy-Desplanques theorem provides a theoretical foundation for maintaining the stability of matrix optimization in high-dimensional spaces. Following this theorem, our proposed gradual mask ensures that the matrix remains strictly diagonally dominant during the optimization process. This guarantees the matrix's invertibility or well-conditioned property and further reduces the mean square error objective function. Our approach consistently improves performance across a wide range of quantization configurations for all models. Notably, the affine transformation method demonstrates great potential for improving performance, particularly for small models and low-bit configurations. In the future, optimizing the affine transformation matrix more effectively deserves careful consideration.

## ACKNOWLEDGMENTS

This work was supported by National Science and Technology Major Project (No. 2022ZD0118202), the National Science Fund for Distinguished Young Scholars (No.62025603), the National Natural Science Foundation of China (No. U21B2037, No. U22B2051, No. 62176222, No. 62176223, No. 62176226, No. 62072386, No. 62072387, No. 62072389, No. 62002305 and No. 62272401), and the Natural Science Foundation of Fujian Province of China (No.2021J01002, No.2022J06001).

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

## A APPENDIX

### A.1 PARETO FRONTS BASED ON WEIGHTED MEMORY

In Figure 4, We show the Pareto frontiers of AffineQuant and OmniQuant based on weighted memory and PPL trade-off for LLaMA1&2 models of different sizes in the 4/4 bit quantization configuration. The results clearly demonstrate that AffineQuant consistently outperforms the current State-Of-The-Art method, OmniQuant, without any additional overhead.

### A.2 STRICTLY DIAGONAL DOMINANCE GUARANTEED

To avoid confusion in the proof between $\alpha$ and the elements in the matrix $A$, we temporarily denote the affine matrix $A$ as $N$.

**Theorem 1** *When the stability factor $\alpha$ is small enough, if $N_e$ is strictly diagonally dominant, then $N_{e+1}$ is strictly diagonally dominant.*

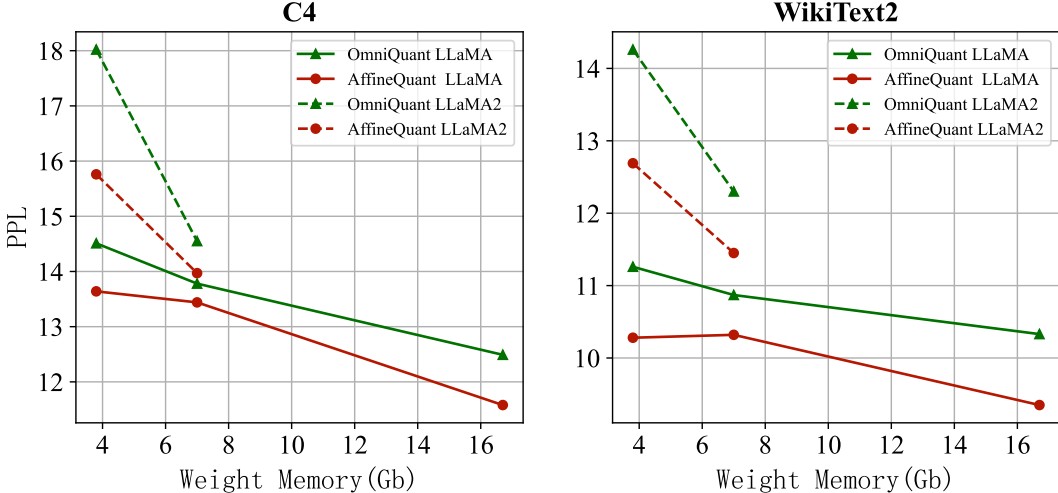

Figure 4: PPL vs. weight-memory Pareto-optimal curves for LLaMA1&2 models of different sizes in the 4/4 bit quantization configuration on C4 and WikiText2.

**Proof 1** *Without loss of generality, we take the $i$-th row of $N_e$. Since $N_e$ is a strictly diagonally dominant matrix, we have,*

$$|n_{ii}^e| > \sum_{j \neq i} |n_{ij}^e|. \tag{10}$$

*Where $n_{ii}^e$, $n_{ij}^e$ are the elements of the epoch $e$ in the $i$-th row, $i$-th column and $i$-th row, $j$-th column of the matrix $N$. According to the above Equation 9, the absolute value of the $i$-th diagonal element of the $e+1$ epoch of matrix $N$ is,*

$$|n_{ii}^{e+1}| = |n_{ii}^e + \eta g_{ii}^e \frac{\partial L^e}{\partial n_{ii}^{e*}}|, \tag{11}$$

$$= |n_{ii}^e + \eta \frac{\partial L^e}{\partial n_{ii}^{e*}}|. \tag{12}$$

*Where $g_{ii}^e = 1$, $n_{ii}^{e*}$ are the $i$-th diagonal elements of $GM$, $N_e^*$ at epoch e, respectively. $L^e$ is the loss at epoch e. Further,*

$$|n_{ii}^{e+1}| = |n_{ii}^0 + \eta \sum_{x=0}^e \frac{\partial L^x}{\partial n_{ii}^{x*}}|. \tag{13}$$

*$n_{ii}^0$ is the scale when the matrix is initialized. Therefore, the diagonal values of the matrix $N$ are not equal to 0 during the optimization process. Next, we focus on the right-hand side of Equation 10 at epoch e+1. Similarly,*

$$\sum_{j \neq i} |n_{ij}^{e+1}| = \sum_{j \neq i} |n_{ij}^e + \eta g_{ii}^e \frac{\partial L^e}{\partial n_{ij}^{e*}}|, \tag{14}$$

$$= \sum_{j \neq i} |n_{ij}^h + \eta \sum_{x=h}^e g_{ii}^x \frac{\partial L^x}{\partial n_{ij}^{x*}}|. \tag{15}$$

*Where $1 \leq h \leq e$ is the epoch at which $n_{ij}$ starts updating. In other words, $n_{ij}^h = 0$, and as h gets smaller $n_{ij}$ gets closer to the diagonal. In addition, $g_{ii}^x = \alpha$. Therefore, we have,*

$$\sum_{j \neq i} |n_{ij}^{e+1}| = \eta \alpha \sum_{j \neq i} |\sum_{x=h}^e \frac{\partial L^x}{\partial n_{ij}^{x*}}|. \tag{16}$$

*To make $\sum_{j \neq i} |n_{ij}^{e+1}| < |n_{ii}^{e+1}|$, we let*

$$\eta \alpha \sum_{j \neq i} |\sum_{x=h}^{e} \frac{\partial L^x}{\partial n_{ij}^{x*}}| < |n_{ii}^0 + \eta \sum_{x=0}^{e} \frac{\partial L^x}{\partial n_{ii}^{x*}}| \tag{17}$$

$$\alpha < \frac{|n_{ii}^0 + \eta \sum_{x=0}^{e} \frac{\partial L^x}{\partial n_{ii}^{x*}}|}{\eta \sum_{j \neq i} |\sum_{x=h}^{e} \frac{\partial L^x}{\partial n_{ij}^{x*}}|} \tag{18}$$

*Thus, when the stability factor $\alpha$ is sufficiently small, if $N_e$ is a strictly diagonally dominant matrix, then $N_{e+1}$ is a strictly diagonally dominant matrix. The theorem is proved.*

□

### A.3 COMPARISON WITH FLEXROUND

Please refer to Table 7.

Table 7: AffineQuant vs. FlexRound. We perform accuracy comparisons on 6 zero-shot tasks.

| | Dataset | PIQA (↑) | ARC-e (↑) | WinoGrande (↑) | BoolQ (↑) | ARC-c (↑) | HellaSwag (↑) | Avg. (↑) |
|---|---|---|---|---|---|---|---|---|
| LLaMA-7B w4a16 | FP16 | 77.37 | 52.52 | 66.85 | 73.12 | 41.38 | 72.99 | 64.04 |
| | FlexRound Lee et al. (2023) | 77.75 | 50.80 | 66.06 | 70.73 | 40.27 | 71.97 | 62.93 |
| | AffineQuant | 77.53 | 51.85 | 66.93 | 70.89 | 38.65 | 71.49 | 62.89 |
| LLaMA-13B w4a16 | FP16 | 79.11 | 59.89 | 70.01 | 68.53 | 44.54 | 76.23 | 66.38 |
| | FlexRound Lee et al. (2023) | 78.78 | 59.55 | 70.40 | 66.39 | 43.77 | 75.52 | 65.73 |
| | AffineQuant | 78.84 | 59.55 | 69.38 | 69.48 | 43.52 | 75.18 | 65.99 |

### A.4 LOSS AND MODEL PERFORMANCE

We maintain consistent matrix initialization while randomly sampling the stability factor $\alpha$, which influences loss convergence, for LLaMA-7B and OPT-6.7B. Using AffineQuant, we obtain the performance of 4/4 bit quantized models based on the sampled solution. In Figure 5, 6, we present scatter plots depicting the output loss of the last transformer block and the corresponding model performance on different datasets. These plots demonstrate a significant positive correlation between loss and model performance, with correlation coefficients of 0.95,0.96 on OPT-6.7B and LLaMA-7B in WikiText2, respectively. Based on this observation, we conclude that the quantization loss of the last transformer block's output exhibits a strong correlation with overall model performance.

### A.5 ADDITIONAL EXPERIMENT

We list additional experiments including:

1. the OPT model on the PTB dataset.
2. OPT model on C4 dataset.
3. LLaMA1&2 on the WikiText2 dataset.

Each experiment included models at different scales and a wide range of quantitative configurations.

### A.6 AFFINE MATRIX

Figure 7 presents a comprehensive collection of affine transformation matrices, encompassing various transformer block locations, training epochs, layers, and quantization configurations. "fc1_Affine_Matrix_A" denotes the affine transformation matrix at fc1, "out_Affine_Matrix_A" represents the affine transformation matrix at out_proj, and "qkv_Affine_Matrix_A" corresponds to the affine transformation matrix at qkv. To ensure consistency, we normalize the matrix values within the range of 0 to 1 using a specified normalization method. Notably, all matrices exhibit the property

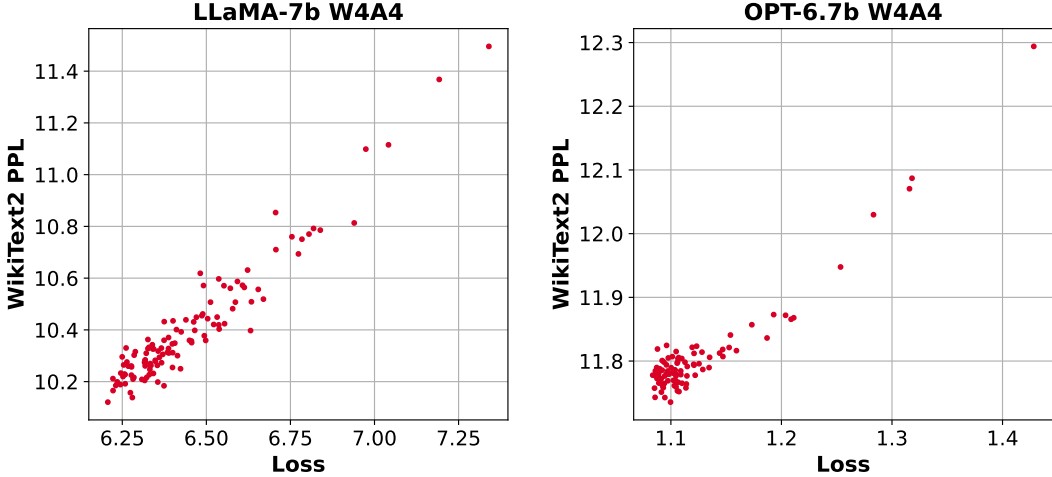

Figure 5: The relationship between WikiText2 PPL and quantization loss of last transformer block on LLaMA-7B and OPT-6.7B with 4/4 bit quantization.

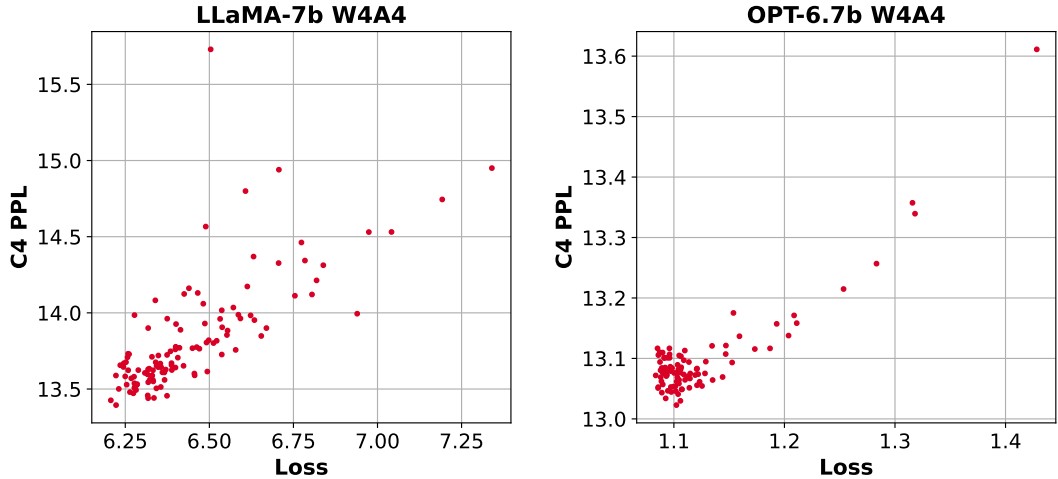

Figure 6: The relationship between C4 PPL and quantization loss of last transformer block on LLaMA-7B and OPT-6.7B with 4/4 bit quantization.

of being strictly diagonally dominant. Additionally, the low-bit affine transformation matrix demonstrates a higher capability to learn rotational features, thereby reducing the model's quantization error compared to the high-bit configuration.

Furthermore, as the training epochs progress, the affine transformation matrix acquires more non-primary diagonal elements. On the other hand, the persistence of an approximate diagonal matrix at high quantization bits elucidates the modest performance improvement observed in high-bit quantization configurations. This phenomenon may also be attributed to the relatively small performance gap between the quantized model and the full-precision model.

## A.7 EXPERIMENTAL DETAILS

To ensure a fair comparison, we align most of our optimization parameters with those of Omni-Quant (Shao et al., 2023). Specifically, we apply INT2, INT3, and INT4 only-weight quantization to OPT (Zhang et al., 2022) and LLaMA1&2 (Touvron et al., 2023a;b) models. Additionally, we employ grouping strategies of 64 or 128 for weight quantization with different bit configurations. The

Table 8: Weight-only quantization PPL(↓) results on the OPT model PTB dataset.

| Config | Method | 125M | 1.3B | 2.7B | 6.7B | 13B | 30B |
|---|---|---|---|---|---|---|---|
| FP16 | - | 32.54 | 16.96 | 15.11 | 13.08 | 12.33 | 11.84 |
| w2a16g128 | RTN | 4.6e3 | 7.1e3 | 2.5e4 | 5.7e3 | 3.0e4 | 6.2e3 |
| | GPTQ (Frantar et al., 2022) | 655.17 | 130.88 | 61.36 | 25.24 | 20.46 | 15.15 |
| | AWQ (Lin et al., 2023) | 263.88 | 71.87 | 43.15 | 19.49 | 17.61 | 14.92 |
| | OmniQuant (Shao et al., 2023) | 126.49 | 34.33 | 25.28 | 18.92 | 16.74 | 14.51 |
| | AffineQuant | 65.23 | 30.06 | 27.11 | 18.22 | 16.35 | 14.09 |
| w2a16g64 | RTN | 5.1e3 | 19.4e3 | 7.7e4 | 6.1e3 | 8.2e3 | 4.1e3 |
| | GPTQ (Frantar et al., 2022) | 245.28 | 55.61 | 36.12 | 19.45 | 17.02 | 14.05 |
| | AWQ (Lin et al., 2023) | 143.18 | 41.19 | 25.08 | 18.00 | 15.83 | 14.92 |
| | OmniQuant (Shao et al., 2023) | 112.10 | 30.36 | 22.63 | 17.58 | 15.70 | 13.98 |
| | AffineQuant | 60.90 | 27.21 | 21.50 | 17.07 | 15.32 | 13.68 |
| w3a16 | RTN | 1.2e3 | 1.1e4 | 1.0e4 | 5.2e3 | 3.6e3 | 1.4e3 |
| | GPTQ (Frantar et al., 2022) | 34.05 | 27.39 | 15.94 | 13.75 | 13.71 | 12.54 |
| | AWQ (Lin et al., 2023) | 80.73 | 33.20 | 224.11 | 18.46 | 35.45 | 66.68 |
| | OmniQuant (Shao et al., 2023) | 40.76 | 19.06 | 16.29 | 13.77 | 12.96 | 12.19 |
| | AffineQuant | 38.38 | 19.14 | 16.32 | 14.19 | 13.54 | 12.48 |
| w3a16g128 | RTN | 64.67 | 222.13 | 337.75 | 39.90 | 65.33 | 34.27 |
| | GPTQ (Frantar et al., 2022) | 45.17 | 19.90 | 17.06 | 14.24 | 12.84 | 12.54 |
| | AWQ (Lin et al., 2023) | 44.07 | 19.59 | 16.52 | 13.98 | 12.87 | 66.68 |
| | OmniQuant (Shao et al., 2023) | 45.29 | 20.42 | 17.08 | 14.23 | 13.49 | 12.54 |
| | AffineQuant | 36.70 | 18.64 | 16.11 | 13.59 | 12.97 | 12.14 |
| w4a16 | RTN | 44.98 | 33.63 | 22.23 | 16.05 | 15.40 | 14.17 |
| | GPTQ (Frantar et al., 2022) | 37.75 | 18.23 | 15.94 | 13.75 | 12.58 | 11.98 |
| | AWQ (Lin et al., 2023) | 38.74 | 18.35 | 15.70 | 13.59 | 12.72 | 12.06 |
| | OmniQuant (Shao et al., 2023) | 34.94 | 17.80 | 15.52 | 13.41 | 12.62 | 11.95 |
| | AffineQuant | 34.29 | 17.55 | 15.49 | 13.30 | 12.54 | 11.97 |
| w4a16g128 | RTN | 36.50 | 33.63 | 22.23 | 16.05 | 15.40 | 14.17 |
| | GPTQ (Frantar et al., 2022) | 35.48 | 17.41 | 15.42 | 13.21 | 12.42 | 11.89 |
| | AWQ (Lin et al., 2023) | 34.95 | 17.46 | 15.33 | 13.28 | 12.46 | 11.90 |
| | OmniQuant (Shao et al., 2023) | 34.28 | 17.40 | 15.28 | 13.25 | 12.46 | 11.94 |
| | AffineQuant | 34.00 | 17.33 | 15.25 | 13.27 | 12.44 | 11.94 |

model's performance is evaluated on the WikiText2 (Merity et al., 2016), PTB (Marcus et al., 1994), and C4 (Raffel et al., 2020) datasets. For algorithm optimization, we randomly select 128 segments from the WikiText2 training set, each containing 2048 tokens, as the calibration dataset. We leverage the scale of SmoothQuant (Xiao et al., 2023) to initialize the diagonal of the affine transformation matrix. As the affine transformation is orthogonal to the translation operation, we incorporate the optimization of the learnable parameter shift and initialize it using Outlier Suppression+ (Wei et al., 2023). Our optimizer, learning rate, epoch, and learnable clipping of quantization parameters are consistent with OmniQuant (Shao et al., 2023). The optimization process is performed on an Nvidia A100 GPU. We conduct a comparative analysis of various weight-only quantization methods, including GPTQ (Frantar et al., 2022), AWQ (Lin et al., 2023), and OmniQuant (Shao et al., 2023).

Table 9: Weight-only quantization PPL(↓) results on the OPT model C4 dataset.

| Config | Method | 125M | 1.3B | 2.7B | 6.7B | 13B | 30B |
|---|---|---|---|---|---|---|---|
| FP16 | - | 24.60 | 14.72 | 13.16 | 11.74 | 11.19 | 10.69 |
| w2a16g128 | RTN | 5.0e3 | 7.7e3 | 3.8e4 | 5.2e3 | 2.8e4 | 6.5e3 |
| | GPTQ (Frantar et al., 2022) | 597.66 | 60.88 | 33.83 | 18.55 | 16.34 | 12.89 |
| | AWQ (Lin et al., 2023) | 168.35 | 38.38 | 26.41 | 16.48 | 14.73 | 12.98 |
| | OmniQuant (Shao et al., 2023) | 80.10 | 27.33 | 21.11 | 16.67 | 14.92 | 13.12 |
| | AffineQuant | 46.22 | 23.28 | 23.10 | 15.62 | 14.60 | 12.93 |
| w2a16g64 | RTN | 3.9e3 | 7.3e3 | 1.2e5 | 6.3e3 | 7.5e3 | 4.0e3 |
| | GPTQ (Frantar et al., 2022) | 133.51 | 31.31 | 23.23 | 16.24 | 14.48 | 12.24 |
| | AWQ (Lin et al., 2023) | 90.19 | 27.34 | 20.01 | 15.20 | 13.90 | 12.43 |
| | OmniQuant (Shao et al., 2023) | 64.01 | 23.71 | 19.16 | 15.44 | 14.16 | 12.80 |
| | AffineQuant | 42.43 | 21.87 | 17.72 | 14.86 | 13.92 | 12.49 |
| w3a16 | RTN | 722.83 | 6.1e3 | 1.2e4 | 5.8e3 | 3.3e3 | 1.4e3 |
| | GPTQ (Frantar et al., 2022) | 37.75 | 19.45 | 13.75 | 15.67 | 12.28 | 11.34 |
| | AWQ (Lin et al., 2023) | 55.73 | 24.56 | 154.49 | 15.84 | 23.71 | 55.01 |
| | OmniQuant (Shao et al., 2023) | 32.17 | 17.10 | 14.93 | 12.78 | 12.13 | 11.37 |
| | AffineQuant | 28.19 | 16.42 | 14.27 | 12.72 | 12.04 | 11.21 |
| w3a16g128 | RTN | 40.13 | 126.47 | 372.23 | 32.56 | 44.12 | 25.70 |
| | GPTQ (Frantar et al., 2022) | 30.08 | 16.47 | 14.54 | 12.48 | 11.58 | 10.91 |
| | AWQ (Lin et al., 2023) | 30.39 | 16.27 | 14.19 | 12.30 | 11.61 | 10.96 |
| | OmniQuant (Shao et al., 2023) | 29.34 | 16.11 | 14.15 | 12.31 | 11.63 | 10.98 |
| | AffineQuant | 27.53 | 16.02 | 13.92 | 12.21 | 11.63 | 10.99 |
| w4a16 | RTN | 31.58 | 24.68 | 17.61 | 13.38 | 12.35 | 11.90 |
| | GPTQ (Frantar et al., 2022) | 27.12 | 15.57 | 13.75 | 12.15 | 11.36 | 10.80 |
| | AWQ (Lin et al., 2023) | 27.64 | 15.65 | 13.71 | 12.04 | 11.42 | 10.83 |
| | OmniQuant (Shao et al., 2023) | 26.36 | 15.28 | 13.58 | 11.97 | 11.41 | 10.80 |
| | AffineQuant | 25.47 | 15.18 | 13.43 | 11.90 | 11.36 | 10.80 |
| w4a16g128 | RTN | 26.79 | 15.71 | 13.79 | 12.31 | 11.51 | 10.94 |
| | GPTQ (Frantar et al., 2022) | 25.96 | 15.05 | 13.40 | 11.87 | 11.26 | 10.74 |
| | AWQ (Lin et al., 2023) | 25.90 | 15.04 | 13.39 | 11.87 | 11.28 | 10.75 |
| | OmniQuant (Shao et al., 2023) | 25.63 | 15.03 | 13.38 | 11.85 | 11.29 | 10.75 |
| | AffineQuant | 25.26 | 14.98 | 13.32 | 11.84 | 11.27 | 10.75 |

Table 10: Weight-only quantization PPL($\downarrow$) results on the LLaMA1&2 model C4 dataset.

| Config | Method | 1-7B | 1-13B | 1-30B | 2-7B | 2-13B |
|--------|--------|------|-------|-------|------|-------|
| FP16 | - | 7.08 | 6.61 | 5.98 | 6.97 | 6.46 |
| w3a16 | RTN | 28.26 | 13.22 | 28.66 | 402.35 | 12.51 |
| | GPTQ (Frantar et al., 2022) | 9.49 | 8.16 | 7.29 | 9.81 | 8.02 |
| | AWQ (Lin et al., 2023) | 13.26 | 9.13 | 12.67 | 23.85 | 13.07 |
| | OmniQuant (Shao et al., 2023) | 8.19 | 7.32 | 6.57 | 8.65 | 7.44 |
| | AffineQuant | 8.03 | 7.20 | 6.55 | 8.57 | 7.56 |
| w3a16g128 | RTN | 8.62 | 7.49 | 6.58 | 8.40 | 7.18 |
| | GPTQ (Frantar et al., 2022) | 7.85 | 7.10 | 6.47 | 7.89 | 7.00 |
| | AWQ (Lin et al., 2023) | 7.92 | 7.07 | 6.37 | 7.84 | 6.94 |
| | OmniQuant (Shao et al., 2023) | 7.34 | 6.76 | 6.11 | 7.35 | 6.65 |
| | AffineQuant | 7.75 | 7.04 | 6.40 | 7.83 | 6.99 |
| w4a16 | RTN | 7.93 | 6.98 | 6.34 | 7.71 | 6.83 |
| | GPTQ (Frantar et al., 2022) | 7.43 | 6.84 | 6.20 | 7.37 | 6.70 |
| | AWQ (Lin et al., 2023) | 7.52 | 6.86 | 6.17 | 7.68 | 6.74 |
| | OmniQuant (Shao et al., 2023) | 7.34 | 6.76 | 6.11 | 7.35 | 6.65 |
| | AffineQuant | 7.30 | 6.75 | 6.10 | 7.29 | 6.64 |
| w4a16g128 | RTN | 7.37 | 6.69 | 6.06 | 7.24 | 6.58 |
| | GPTQ (Frantar et al., 2022) | 7.21 | 6.69 | 6.06 | 7.12 | 6.56 |
| | AWQ (Lin et al., 2023) | 7.21 | 6.70 | 6.05 | 7.13 | 6.56 |
| | OmniQuant (Shao et al., 2023) | 7.21 | 6.69 | 6.06 | 7.12 | 6.56 |
| | AffineQuant | 7.20 | 6.69 | 6.05 | 7.12 | 6.56 |

Table 11: Weight-only quantization PPL(↓) results on the LLaMA1&2 model WikiText2 dataset.

| Config | Method | 1-7B | 1-13B | 1-30B | 2-7B | 2-13B |
|---|---|---|---|---|---|---|
| FP16 | - | 5.68 | 5.09 | 4.10 | 5.47 | 4.88 |
| w2a16 | RTN | 1.1e5 | 6.8e4 | 2.4e4 | 3.8e4 | 5.6e4 |
| | GPTQ (Frantar et al., 2022) | 2.1e3 | 5.5e3 | 499.75 | 7.7e3 | 2.1e3 |
| | OmniQuant (Shao et al., 2023) | 15.47 | 13.21 | 8.71 | 37.37 | 17.21 |
| | AffineQuant | 9.53 | 7.54 | 8.35 | 35.07 | 12.42 |
| w2a16g128 | RTN | 1.9e3 | 781.20 | 68.04 | 4.2e3 | 122.08 |
| | GPTQ (Frantar et al., 2022) | 44.01 | 15.60 | 10.92 | 36.77 | 28.14 |
| | AWQ (Lin et al., 2023) | 2.6e5 | 2.8e5 | 2.4e5 | 2.2e5 | 1.2e5 |
| | OmniQuant (Shao et al., 2023) | 10.53 | 8.37 | 7.77 | 12.84 | 9.15 |
| | AffineQuant | 13.51 | 7.22 | 6.49 | 10.87 | 7.64 |
| w2a16g64 | RTN | 188.32 | 101.87 | 19.20 | 431.97 | 26.22 |
| | GPTQ (Frantar et al., 2022) | 22.10 | 10.06 | 8.54 | 20.85 | 22.44 |
| | AWQ (Lin et al., 2023) | 2.5e5 | 2.7e5 | 2.3e5 | 2.1e5 | 1.2e5 |
| | OmniQuant (Shao et al., 2023) | 9.41 | 7.62 | 7.14 | 10.56 | 8.14 |
| | AffineQuant | 8.35 | 6.98 | 6.20 | 9.05 | 7.11 |
| w3a16 | RTN | 25.73 | 11.39 | 14.95 | 539.48 | 10.68 |
| | GPTQ (Frantar et al., 2022) | 8.06 | 6.76 | 5.84 | 8.37 | 6.44 |
| | AWQ (Lin et al., 2023) | 11.88 | 7.45 | 10.07 | 24.00 | 10.45 |
| | OmniQuant (Shao et al., 2023) | 6.49 | 5.68 | 4.74 | 6.58 | 5.58 |
| | AffineQuant | 6.30 | 5.60 | 4.68 | 6.55 | 5.62 |
| w3a16g128 | RTN | 7.01 | 5.88 | 4.87 | 6.66 | 5.51 |
| | GPTQ (Frantar et al., 2022) | 6.55 | 5.62 | 4.80 | 6.29 | 5.42 |
| | AWQ (Lin et al., 2023) | 6.46 | 5.51 | 4.63 | 6.24 | 5.32 |
| | OmniQuant (Shao et al., 2023) | 6.15 | 5.44 | 4.56 | 6.03 | 5.28 |
| | AffineQuant | 6.14 | 5.45 | 4.59 | 6.08 | 5.28 |
| w4a16 | RTN | 6.43 | 5.55 | 4.57 | 6.11 | 5.20 |
| | GPTQ (Frantar et al., 2022) | 6.13 | 5.40 | 4.48 | 5.83 | 5.13 |
| | AWQ (Lin et al., 2023) | 6.08 | 5.34 | 4.39 | 6.15 | 5.12 |
| | OmniQuant (Shao et al., 2023) | 5.86 | 5.21 | 4.25 | 5.74 | 5.02 |
| | AffineQuant | 5.84 | 5.20 | 4.23 | 5.69 | 5.01 |
| w4a16g128 | RTN | 5.96 | 5.25 | 4.23 | 5.72 | 4.98 |
| | GPTQ (Frantar et al., 2022) | 5.85 | 5.20 | 4.23 | 5.61 | 4.98 |
| | AWQ (Lin et al., 2023) | 5.81 | 5.20 | 4.21 | 5.62 | 4.97 |
| | OmniQuant (Shao et al., 2023) | 5.77 | 5.17 | 4.19 | 5.58 | 4.95 |
| | AffineQuant | 5.77 | 5.17 | 4.19 | 5.58 | 4.95 |

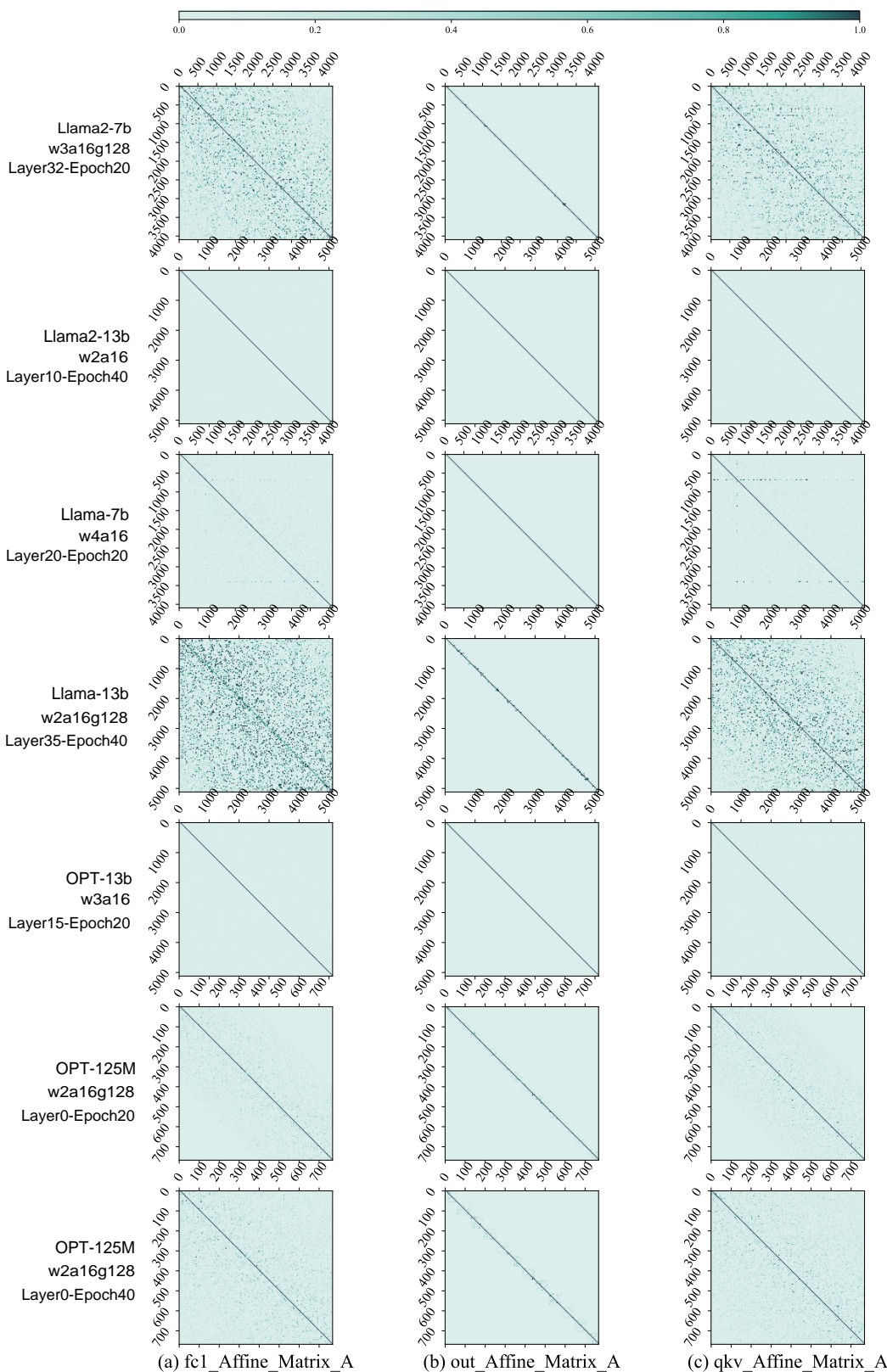

(a) fc1_Affine_Matrix_A      (b) out_Affine_Matrix_A      (c) qkv_Affine_Matrix_A

Figure 7: Affine transformation matrix for different quantization configurations, different layers, and different training epochs for OPT and LLaMA1&2.

