# OpenReview forum: "AffineQuant: Affine Transformation Quantization for Large Language Models"
_ICLR.cc/2024/Conference — ICLR 2024 poster_

### Official Review · Reviewer_MkcQ · 2023-10-30

**Soundness:** 3 good
**Presentation:** 3 good
**Contribution:** 3 good
**Rating:** 8
**Confidence:** 5

**Summary:**

This work develops a quantization method that uses an affine matrix transformation to quantize the weights of a transformer. By doing so, the objective function has more parameters to be optimized and can achieve lower quantization error when the weight matrix is compressed to a low-bit representation (2-bit).

Recommendation: The presentation and results are currently misleading. I might miss something, but currently, the method has no practical benefit. If the authors can explain to me what I am missing, I am happy to raise my score significantly (2-4 points). I will also raise my score by 2 points if the presentation of 2-bit results is dropped in favor of an analysis of information density (scaling curves that show Pareto fronts in terms of performance per bit of total memory footprint).

**Strengths:**

The results from affine quantization are strong compared to baselines.

**Weaknesses:**

- The paper misleadingly highlights 2-bit quantization when the same paper shows higher information density for 4-bit quantization (4-bit + 7B params > 2-bit + 13B params, etc.). This is a mistake that is propagated in the literature. I will not accept work that presents quantization results like this because it is very misleading.
- It is unclear what the benefit of the method is. To use the quantization in practice, the inverse affine transformation matrix is needed to convert the input tensor. However, by having to store this matrix, the size of the model is doubled, which negates FLOPs and memory benefits. Am I missing something?

**Questions:**

- How does the process work at inference time? If you optimize with the affine matrix during quantization, you surely need it during inference time.
- How fast is inference for this method compared to baselines?

---

> ### Author Response · Authors · 2023-11-22
>
> **W1: When the paper shows that 4-bit quantization has a higher information density, the results emphasizing 2-bit quantization are misleading.**
>
> **Reply:** We admit the conclusion that 4 bit quantization has a higher information density than 2 bit quantization. To prevent any potential misinterpretation, we have removed references to the 2 bit quantization results from the abstract and the body of the paper, replacing them with the 4/4 bit results that incur no additional overhead. Our experimental results continue to outperform after modifying the quantization configuration.
>
> **W2: In practice it is necessary to store affine matrices, which doubles the model size and thus negates the FLOPs and memory benefits of quantization.**
>
> **Reply:** The modified AffineQuant doesn't need to store matrices because it can be fused into the weight and bias of LayerNorm and Linear. Specifically, we optimize only the diagonal elements of the affine matrix, enabling integration with the weight and bias of LayerNorm. Additionally, we apply the affine matrix between v and out\_proj, as well as between q,k output activations, allowing fusion with the weight and bias of the q, k, v, out_proj Linear layer. We take the linear layer v and the previous LayerNorm as an example and formally,
> $$
> \begin{align}
> \gamma^{'}&=\gamma\cdot diag(A_{qkv})^{-1}, \tag 1 \\\\
> \beta^{'}&=\beta\cdot diag(A_{qkv})^{-1}, \tag 2 \\\\
> W_{v}^{'} &= diag(A_{qkv})W_{v}A_{vo}^{-1}, \tag 3 \\\\
> bias_{v}^{'} &= bias_{v}A_{vo}^{-1}. \tag 4
> \end{align}
> $$
> Where $\gamma$, ${\gamma}^{'}$, $\beta$, ${\beta}^{'}$ are the LayerNorm parameters before and after the affine transformation. $diag(\cdot)$ is the diagonal matrix fetching operation. $A_{qkv}$ is the optimal affine matrix at the Linear layer $q$,$k$,$v$. $W_v$, $bias_v$ are the weights and bias of the Linear layer $v$. $A_{vo}^{-1}$ is the inverse of the optimal affine matrix between $v$ and out projection.
>
> Therefore, AffineQuant does not incur additional FLOPs as well as memory overhead. Meanwhile, we compare the 4/4 bit quantization performance of LLaMA1&2 models on WikiText2 and C4 datasets in the following table.
>
> |            | Methods     | WikiText2 | C4    |
> | ---------- | ----------- | --------- | ----- |
> | LLaMA-7B   | OmniQuant   | 11.26     | 14.51 |
> |            | AffineQuant | 10.28     | 13.64 |
> | LLaMA-13B  | OmniQuant   | 10.87     | 13.78 |
> |            | AffineQuant | 10.32     | 13.44 |
> | LLaMA-30B  | OmniQuant   | 10.33     | 12.49 |
> |            | AffineQuant | 9.35      | 11.58 |
> | LLaMA2-7B  | OmniQuant   | 14.26     | 18.02 |
> |            | AffineQuant | 12.69     | 15.76 |
> | LLaMA2-13B | OmniQuant   | 12.30     | 14.55 |
> |            | AffineQuant | 11.45     | 13.97 |
>
> **Q1-2: How does the process work at inference time? How fast is inference for this method compared to baselines?**
>
> **Reply:** For the experiments in the table above we optimize the matrix after LayerNorm for diagonal elements only. This allows us to fuse the affine matrix with the LayerNorm weights and bias. For the remaining positions, we apply matrix transformations. After optimization, the matrix can be further fused with the weights and bias of the preceding linear layer. Consequently, AffineQuant can be achieved without introducing any additional overhead to model inference. We utilize the MLC-LLM [2] library to compile the LLaMA model at various scales under only-weight quantization configuration on a single A100, and the resulting inference speeds are presented in the following table.
>
> | LLaMA     | 7B    | 7B    | 7B      | 13B   | 13B   | 13B     | 30B   | 30B   | 30B     | 65B   | 65B   | 65B     |
> | --------- | ----- | ----- | ------- | ----- | ----- | ------- | ----- | ----- | ------- | ----- | ----- | ------- |
> |           | WM    | RM    | token/s | WM    | RM    | token/s | WM    | RM    | token/s | WM    | RM    | token/s |
> | FP        | 12.6G | 14.4G | 69.2    | 24.3G | 27.1G | 52.5    | 60.6G | 66.1G | 23.9    | OOM   | -     | -       |
> | W4A16g128 | 3.8G  | 5.7G  | 155.3   | 7.0G  | 10.0G | 102.7   | 16.7G | 21.7G | 45.2    | 33.0G | 41.0G | 25.2    |
> | W3A16g128 | 3.2G  | 5.1G  | 87.9    | 5.8G  | 8.7G  | 63.7    | 13.7G | 18.7G | 30.3    | 27.0G | 35.1G | 15.7    |
> | W2A16g128 | 2.2G  | 4.1G  | 84.5    | 4.0G  | 7.5G  | 95.2    | 9.2G  | 14.1G | 38.3    | 18.0G | 25.6G | 25.6    |
>
> **Q3: scaling curves that show Pareto fronts in terms of performance per bit of total memory footprint.**
>
> **Reply:** In the Appendix, we provide the PPL vs. weight-memory Pareto-optimal curves for LLaMA1&2 models of different sizes in the 4/4 bit quantization configuration. The results clearly demonstrate that AffineQuant consistently outperforms the current State-Of-The-Art method, OmniQuant, without any additional overhead.

---

> > ### Comment · Reviewer_MkcQ · 2023-11-23
> > **Thank you for helping me understand.**
> >
> > Thank you for your rebuttal, this was very helpful. I think what was confusing me is the term "fused". Fused operations often load memory in SRAM and are reused. As such, they have no memory and FLOPs advantage. I think a better term for your method would be "merged" instead of "fused".
> >
> > Otherwise, the removal of the reference to 2-bit quantization make all claims robust. As such, I no longer see any problem with the paper.
> >
> > Since everything can be merged into layernorms and other layers this method is actually quite practical and shows strong results. As such, all my concerns are alleviated and I am impressed. I would like this paper to be accepted. I would be happy if this paper would be highlighted as a spotlight.

---

> > > ### Author Response · Authors · 2023-11-23
> > >
> > > We are very grateful for your recognition and comments, and it is these suggestions that motivate us to keep improving our manuscript. We will replace 'fused' with 'merged' and refine other aspects of our manuscript's expression.

---

### Official Review · Reviewer_E6YB · 2023-10-31

**Soundness:** 3 good
**Presentation:** 3 good
**Contribution:** 2 fair
**Rating:** 5
**Confidence:** 4

**Summary:**

The paper introduces "AffineQuant", a method to optimize Large-scale Language Models (LLMs) using equivalent Affine transformations in Post-Training Quantization (PTQ). Traditional PTQ techniques often resulted in significant errors, especially in low-bit configurations. AffineQuant expands the optimization scope, reducing these errors. Using a unique gradual mask optimization method aligned with the Levy-Desplanques theorem, invertibility of transformations is ensured. The results show AffineQuant outperforms existing methods, especially in low-bit configurations, making it a promising tool for model compression and deployment.

**Strengths:**

- This paper is well-written and successfully exhibit their features.
- This paper extends current PTQ papers to introduce affine transformation to MRE (minimum reconstruction error) methods. It could be reasonable in the line of quantization papers' history.

**Weaknesses:**

- The paper appears to overlook contemporary quantization methodologies like FlexRound. Notably, FlexRound seems to be a successor in the lineage of MRE-PTQ techniques, such as BrecQ and QDrop, designed for LLM compression. A comparative analysis with these methods would accentuate the novelty and efficacy of the proposed technique. Reference for consideration: https://arxiv.org/abs/2306.00317.

- Regarding Figure 3, my understanding is that the loss of a partial layer during MRE PTQ might not directly correlate with improved model performance, especially in the context of generative AI models.

- I have a few observations concerning Section 4, which details experimental results:
   1) A significant concern is the sole reliance on PPL scores as a performance metric. While it corresponds to dataset loss, recent LLM compression research suggests that PPL may not fully capture the generative capabilities post-quantization. It might be worthwhile to consider metrics like common-sense reasoning or evaluations using the MMLU dataset.
   2) Additionally, an increase in PPL scores by over 10 points indicates a significant degradation in the generation capabilities of the quantized model. Thus, contrasting the performance of such models, like the W2A16 results mentioned in the abstract, might not provide meaningful insights. A closer look at the actual generation outputs from these severely quantized models could reveal issues like non-coherent results or repeated verbiage.
Lastly, larger models could yield different outcomes, as they might exhibit distinct patterns of outliers or weight distributions.

**Questions:**

included in weaknesses.

---

> ### Author Response · Authors · 2023-11-22
>
> **W1: Comparisons and analyses with FlexRound serve to emphasize AffineQuant's novelty and validity.**
>
> **Reply:** FlexRound [1] is a commendable paper. However, it is orthogonal to our approach. Because,
>
> 1. FlexRound proposes an element-by-element division weight rounding scheme designed to break away from the upper or lower rounding of Adaround [2], BRECQ [3]. AffineQuant proposes an affine transformation matrix to make the distribution of weights and activations more suitable for quantization.
> 2. FlexRound's rounding scheme can explore more aggressive rounding scenarios for a large magnitude of weight values to achieve the global reconstruction error optimization. AffineQuant proposes the gradual mask to provide stable computation of the inverse of the affine matrix during the optimization process.
> 3. The optimized weight rounding values are orthogonal to the optimized weight activation distribution. After using AffineQuant to fuse the optimal affine matrix into the original model, we can use FlexRound to minimize the reconstruction error by performing secondary reconstruction with the optimized model weight rounding values.
>
> AffineQuant focuses on post-training quantization without fine-tuning, making the LoRA fine-tuning experiments in FlexRound not applicable to our approach. Furthermore, FlexRound does not provide clear information on whether it quantizes the input activations of both matmul operations for weight-activation quantization in LLM. In contrast, AffineQuant quantizes all input activations except for the softmax input activation of the second matmul. To compare their performance, we present the results of FlexRound in the 4/16 bit quantization configuration in the table below. It is important to note that AffineQuant, as shown in the table, only updates the diagonal elements of the matrix after LayerNorm to maintain model inference without any additional overhead. We place the comparison experiments in Appendix of the paper.
>
> |                        | PIQA($\uparrow$) | ARC-e($\uparrow$) | WinoGrande($\uparrow$) | BoolQ($\uparrow$) | ARC-c($\uparrow$) | HellaSwag($\uparrow$) | Avg.($\uparrow$) |
> | ---------------------- | ---------------- | ----------------- | ---------------------- | ----------------- | ----------------- | --------------------- | ---------------- |
> | FP16                   | 77.37            | 52.52             | 66.85                  | 73.12             | 41.38             | 72.99                 | 64.04            |
> | LLaMA-7B, FlexRound    | 77.75            | 50.80             | 66.06                  | 70.73             | 40.27             | 71.97                 | 62.93            |
> | LLaMA-7B, AffineQuant  | 77.53            | 51.85             | 66.93                  | 70.89             | 38.65             | 71.49                 | 62.89            |
> | FP16                   | 79.11            | 59.89             | 70.01                  | 68.53             | 44.54             | 76.23                 | 66.38            |
> | LLaMA-13B, FlexRound   | 78.78            | 59.55             | 70.40                  | 66.39             | 43.77             | 75.52                 | 65.73            |
> | LLaMA-13B, AffineQuant | 78.84            | 59.55             | 69.38                  | 69.48             | 43.52             | 75.18                 | 65.99            |
>
> **W2: Loss of some layers during MRE may not be directly related to improved model performance.**
>
> **Reply:** You raise a very good question. We have done sampling experiments based on this question. Specifically, we maintain consistent matrix initialization while randomly sampling the stability factor $\alpha$, which influences loss convergence, for LLaMA-7B and OPT-6.7B. Using AffineQuant, we obtain the performance of 4/4 bit quantized models based on the sampled solution. In the Appendix of our paper, we present scatter plots depicting the output loss of the last transformer block and the corresponding model performance. These plots demonstrate a significant positive correlation between loss and model performance, with correlation coefficients of 0.95,0.96 on OPT-6.7B and LLaMA-7B, respectively. Based on this observation, we conclude that the quantization loss of the last transformer block's output exhibits a strong correlation with overall model performance. Thanks again for your constructive comments.

---

> > ### Author Response · Authors · 2023-11-22
> >
> > **W3.1: Rely solely on PPL scores as a performance metric.**
> >
> > **Reply:** We evaluate the quantization performance of LLaMA-7B, 13B, 30B on six zero-shot datasets using 4/4 bit quantization in the following table. In addition, we present few-shot MMLU (k=5) 4/4 bit quantization results on LLaMA-7B, 13B, 30B and LLaMA2-7B, 13B models of AffineQuant. As we utilize OmniQuant to evaluate the MMLU display metrics for Nan, further tuning of the MMLU results obtained from OmniQuant is required. And the final conclusion on the above datasets is similar to the PPL score. Replacement metrics comparison experiments are placed in the Appendix of the paper.
> >
> > |                        | PIQA($\uparrow$) | ARC-e($\uparrow$) | WinoGrande($\uparrow$) | BoolQ($\uparrow$) | ARC-c($\uparrow$) | HellaSwag($\uparrow$) | Avg.($\uparrow$) |
> > | ---------------------- | ---------------- | ----------------- | ---------------------- | ----------------- | ----------------- | --------------------- | ---------------- |
> > | LLaMA-7B, OmniQuant    | 66.15            | 45.20             | 53.43                  | 63.51             | 31.14             | 56.44                 | 52.65            |
> > | LLaMA-7B, AffineQuant  | 69.37            | 42.55             | 55.33                  | 63.73             | 31.91             | 57.65                 | 53.42            |
> > | LLaMA-13B, OmniQuant   | 69.69            | 47.39             | 55.80                  | 62.84             | 33.10             | 58.96                 | 54.37            |
> > | LLaMA-13B, AffineQuant | 66.32            | 43.90             | 54.70                  | 64.10             | 29.61             | 56.88                 | 52.58            |
> > | LLaMA-30B, OmniQuant   | 71.21            | 49.45             | 59.19                  | 65.33             | 34.47             | 64.65                 | 56.63            |
> > | LLaMA-30B, AffineQuant | 70.84            | 49.41             | 58.64                  | 70.12             | 37.12             | 65.53                 | 58.61            |
> >
> > |                         | W/A  | MMLU($\uparrow$) |
> > | :---------------------- | ---- | ---------------- |
> > | LLaMA-7B, AffineQuant   | 4/4  | 27.27            |
> > | LLaMA-13B, AffineQuant  | 4/4  | 28.29            |
> > | LLaMA-30B, AffineQuant  | 4/4  | 32.62            |
> > | LLaMA2-7B, AffineQuant  | 4/4  | 27.83            |
> > | LLaMA2-13B, AffineQuant | 4/4  | 29.90            |
> >
> > **W3.2: Comparing the performance of very low bit quantization models may not provide meaningful insights.**
> >
> > **Reply:** We acknowledge and appreciate your statement. In response to your comments, we have examined the 2-bit generation results and identified issues such as duplicate outputs. We have made adjustments to the evaluation metrics and modified the configurations for low-bit quantization, as indicated in the table above. Despite these changes, the conclusions remain consistent. Furthermore, we have replaced the experiments in the abstract that emphasized the 2 bit results with a section focusing on the 4/4 bit results.
> >
> >
> >
> > [1] Lee et al. FlexRound: Learnable Rounding based on Element-wise Division for Post-Training Quantization. ICML 2023
> >
> > [2] Nagel et al. Up or Down? Adaptive Rounding for Post-Training Quantization. ICML 2020
> >
> > [3] Li et al. Brecq: Pushing the limit of post-training quantization by block reconstruction. ICLR 2021

---

> ### Author Response · Authors · 2023-11-23
>
> Thanks again for your great efforts and constructive advice in reviewing this paper! With the discussion period drawing to a close, we expect your feedback and thoughts on our reply. We put a significant effort into our response, with several new experiments and discussions. We sincerely hope you can consider our reply in your assessment. We look forward to hearing from you, and we can further address unclear explanations and remaining concerns if any.
>
> Regards,
>
> Authors

---

> > ### Comment · Reviewer_E6YB · 2023-11-23
> >
> > Thank you for the detailed response. I will revise my score since the author has acknowledged and made corrections regarding the part I thought was the most problematic (the PPL results for 2-bit).

---

> > > ### Author Response · Authors · 2023-11-23
> > >
> > > Thank you for your valuable comments. They will assist in refining and rationalizing our experiments.

---

### Official Review · Reviewer_5dxF · 2023-10-31

**Soundness:** 4 excellent
**Presentation:** 3 good
**Contribution:** 4 excellent
**Rating:** 8
**Confidence:** 3

**Summary:**

Propose affine transformation in PTQ (in line with "equivalent transformations" line of work) based on specialized optimization approach involving a “gradual masking” to ensure a viable affine matrix is trained.

**Strengths:**

- A more general approach than several proceedings works in the space of “equivalent transformations”, and a nice explanation in Section 3.1
- strong empirical results

**Weaknesses:**

- I imagine there is an increased computational cost to the proposed method. Is that the reason why results on larger OPT (ie 66B) or Llama (70B) models were not reported? My understanding is that this is the main tradeoff: a more powerful quantization method, but at an increased computational cost?

**Questions:**

- In terms of a computation vs quantization performance tradeoff, what do the authors think about just fine-tuning the model after some standard quantization method? I understand how the gradual mask training approach makes this cheaper than fine-tuning the full model, but it still appears expensive enough that it’s difficult to get results on the largest OPT/Llama models. I think this is an interesting method, more of a point of discussion.

---

> ### Author Response · Authors · 2023-11-22
>
> **W1: Is the increased computational cost the reason for not reporting the results of the larger model?**
>
> **Reply:** For OPT-66B and LLaMA-65B models the optimization time is 82h and 78h respectively, which is much reduced compared to LLM-QAT [1] (80 GPU hours of fine-tuning time on LLaMA-7B). The ppl of OPT-66B and LLaMA-65B on WikiText2 are 10.19 and 8.85, respectively. we will report the larger model runtime comparison and the model performance in a future version.
>
> **Q1: What do the authors think about just fine-tuning the model after some standard quantization method in terms of a computation vs quantization performance tradeoff?**
>
> **Reply:** AffineQuant methods, based on post-training quantization, offer significant efficiency and performance advantages over fine-tuning methods. This holds true for both training and inference overhead.
>
> 1. In terms of training costs, **Q**uantization **A**ware-**T**raining (QAT) for LLM involves the fine-tuning process to address quantization errors. LLM-QAT [1] requires 80 GPU hours to fine-tune the model for a single A100 on LLaMA-7B. The time taken to generate the data used for fine-tuning the model on LLaMA-7B is not included in this count. In comparison, AffineQuant quantizes LLaMA-7B in just 6.5 hours on a single A100. Similarly, for LLaMA-65B, AffineQuant takes only 78 hours to obtain the quantized model, which is comparable to LLM-QAT on LLaMA-7B. However, LLM-QAT does not report the training cost of the 65B model. Despite the significantly expensive fine-tuning process, LLM-QAT does not demonstrate any advantage over AffineQuant in terms of experimental results. Please refer to the table below for further details.
>
> |                     | Bits  | PIQA($\uparrow$) | ARC-e($\uparrow$) | WinoGrande($\uparrow$) | BoolQ($\uparrow$) | ARC-c($\uparrow$) | HellaSwag($\uparrow$) | Avg.($\uparrow$) |
> | ------------------- | ----- | ---------------- | ----------------- | ---------------------- | ----------------- | ----------------- | --------------------- | ---------------- |
> | FP16                | -     | 79.3             | 73.0              | 70.0                   | 76.8              | 48.0              | 76.1                  | 70.5             |
> | LLM-QAT+SmoothQuant | 4-4-4 | 55.9             | 35.5              | 50.6                   | 62.4              | 26.4              | 47.8                  | 46.4             |
> | AffineQuant         | 4-4-4 | 69.4             | 42.6              | 55.3                   | 63.7              | 31.9              | 57.7                  | 53.4             |
>
> 2. For the inference cost, for the experiments in the table above we optimize the matrix after LayerNorm for diagonal elements only. This allows us to fuse the affine matrix with the LayerNorm weights and bias. For the remaining positions, we apply matrix transformations. After optimization, the matrix can be further fused with the weights and bias of the preceding linear layer. Consequently, AffineQuant can be achieved without introducing any additional overhead to model inference. We utilize the MLC-LLM [2] library to compile the LLaMA model at various scales under only-weight quantization configuration on a single A100, and the resulting inference speeds are presented in the following table.
>
> | LLaMA     | 7B    | 7B    | 7B      | 13B   | 13B   | 13B     | 30B   | 30B   | 30B     | 65B   | 65B   | 65B     |
> | --------- | ----- | ----- | ------- | ----- | ----- | ------- | ----- | ----- | ------- | ----- | ----- | ------- |
> |           | WM    | RM    | token/s | WM    | RM    | token/s | WM    | RM    | token/s | WM    | RM    | token/s |
> | FP        | 12.6G | 14.4G | 69.2    | 24.3G | 27.1G | 52.5    | 60.6G | 66.1G | 23.9    | OOM   | -     | -       |
> | W4A16g128 | 3.8G  | 5.7G  | 155.3   | 7.0G  | 10.0G | 102.7   | 16.7G | 21.7G | 45.2    | 33.0G | 41.0G | 25.2    |
> | W3A16g128 | 3.2G  | 5.1G  | 87.9    | 5.8G  | 8.7G  | 63.7    | 13.7G | 18.7G | 30.3    | 27.0G | 35.1G | 15.7    |
> | W2A16g128 | 2.2G  | 4.1G  | 84.5    | 4.0G  | 7.5G  | 95.2    | 9.2G  | 14.1G | 38.3    | 18.0G | 25.6G | 25.6    |
>
>
>
> [1] Liu et al., LLM-QAT: Data-Free Quantization Aware Training for Large Language Models.
>
> [2] MLC-LLM, https://github.com/mlc-ai/mlc-llm.

---

> > ### Comment · Reviewer_5dxF · 2023-11-23
> > **Response**
> >
> > Thanks for the interesting followup! The comparison to LLM-QAT is informative, this is certainly a regime where there's a significant gap between fine-tuning and this proposed method.

---

> > > ### Author Response · Authors · 2023-11-23
> > >
> > > Thank you for acknowledging our response and efforts! We're glad to hear that you're satisfied.

---

### Official Review · Reviewer_gGYu · 2023-11-01

**Soundness:** 3 good
**Presentation:** 3 good
**Contribution:** 3 good
**Rating:** 8
**Confidence:** 4

**Summary:**

The paper introduces a method for post-training quantization of Large Language Models (LLMs), specifically focusing on the linear layers. This quantization is uniform, employing fixed bit widths across the layers. The proposed approach involves learning an affine transformation (which is more general than single scaling) of weights before quantization by minimizing the MSE loss. The authors aim to ensure that this matrix remains non-singular, employing the Levy-Desplanques theorem and their proposed Gradual Mask (GM) method. Experiments are conducted on various model variants, including OPT and LLaMA, where the proposed algorithm is comparable or outperforms previous works, especially in low bit-width configurations.

=================================

Update:
The authors addressed my concerns, and I have increased the given rating correspondingly.

**Strengths:**

1. The paper expands on pre-quantization transformations by introducing invertible affine transformations, which are more versatile than previous methods that rely on simple scaling.
2. The proposed post-training quantization method for LLMs addresses a timely and pressing need due to the growing popularity of LLMs with their large model sizes.
3. The method's ability to avoid model retraining is a substantial practical advantage since retraining LLMs can be computationally expensive and time-consuming.
4. The weight transformation before quantization is not limited to LLMs; it can be applied to linear layers in different models as well.
5. Inference speed is unaffected, as the additional matrices can be fused with the weight matrices.
6. The paper provides extensive experimental results, including comparisons across different datasets and NLP models.

**Weaknesses:**

1. A significant weakness of this paper is the lack of clarity in explaining the implementation of the core concept, which involves the use of strictly diagonal matrices and the proposed Gradual Mask (GM). Figure 2 suggests that the GM matrix is element-wise multiplied by the matrix A, but the description implies a different interpretation, where it functions as a learning rate for each element in A. This discrepancy needs further clarification to provide a complete understanding of the method.

2. The hyper-parameters $b$ (bit-width) and $\alpha$ (stability factor) may introduce significant computational overhead in the pursuit of determining the optimal trade-off between model size and accuracy.

**Questions:**

1. The method employs non-differentiable components in its optimization approach. It would be interesting to understand how the authors address this challenge in practice.

---

> ### Author Response · Authors · 2023-11-22
>
> **W1: The lack of clarity in explaining the implementation of the core concept.**
>
> **Reply:**
>
> 1. We apologize for the lack of clarity in the presentation of the concept. As you mentioned, we have modified our description in the newest version:
> 2. Gradual Mask (GM) is a learning rate regulator that achieves its purpose by element-wise dot-producting with the matrix A.
> 3. Specifically, the impact of the GM matrix on the optimization process can be divided into two aspects. Here, we present the optimization process for the matrix A after incorporating the GM.
>
> $$
> \begin{align}
> 	 \textbf{Forward:} \quad & A^{*}_{e} = A\_{e} \circ GM\_{e}, \tag 1\\\\
> 	 \textbf{Backward:} \quad & A\_{e+1} = A\_{e} + \eta \\frac{\\partial L}{\\partial A^{\*}\_{e}} \\frac{\\partial A^{\*}\_{e}}{\\partial A\_{e}}, \tag 2\\\\
>   &\quad\quad~=A\_{e}+\eta GM\_{e}\\frac{\\partial L}{\\partial A^{\*}\_{e}}. \tag 3
> \end{align}
> $$
>
> ​	Where $\circ$ is the Hadamard product. $A_{e}$ and $GM_{e}$ are the matrices $A$ and Gradual Mask (GM) matrix in epoch $e$, respectively. $\eta$ is the learning rate of matrix $A$. $L$ is the optimization loss. The GM matrix effectively reduces the magnitude of non-principal diagonal elements in matrix $A$ during forward propagation when the stability factor $\alpha$ is less than $1$. This ensures the existence of a stable inverse matrix of $A^{*}$ in the optimization process during epoch $e$, as per the Levy-Desplanques theorem. In backward propagation, GM affects the learning rate $\eta$, thereby suppressing the update rate of non-primary diagonal elements in matrix $A$. Consequently, the impact of GM on $\eta$ ensures that matrix $A$ in epoch $e+1$ maintains strictly diagonally dominant, satisfying the Levy-Desplanques theorem.
>
> **W2: The hyper-parameters $b$ (bit-width) and $\alpha$ (stability factor) may introduce computational overhead.**
>
> **Reply:** Both the hyperparameter $b$ (bit-width) and $\alpha$ are determined by the requirements in different application scenarios. We thus conclude that, neither hyperparameter is chosen in a way that introduces additional overhead to the optimization process.
>
> **Q1: How the authors address the non-differentiable components in its optimization approach.**
>
> **Reply:** The gradient approximation is determined from **S**traight-**T**hrough **E**stimator (STE) [1], which is formally represented by the following equation:
> $$
> \frac{\partial L}{\partial x} = \frac{\partial L}{\partial Q(x)}\frac{\partial Q(x)}{\partial x}\approx \frac{\partial L}{\partial Q(x)}\textbf{1}_{x},\tag 4
> $$
>
> $$
> \textbf{1}_{x} = \\left\\{
>     \begin{array}{ll}
>         1 & 0\leq \lfloor \frac{x}{\Delta} \rceil+zp \leq 2^n-1, \\\\
>         0 & otherwise. \\\\
>     \end{array}
> \right\. \tag 5
> $$
>
> Please kindly refer to the `round_ste` function in the anonymous link `quantize/quantizer.py` for the precise code implementation.
>
> [1] Bengio~et al., Estimating or propagating gradients through stochastic neurons for conditional computation.

---

> > ### Comment · Reviewer_gGYu · 2023-11-22
> >
> > Are there any guarantees that if matrix $A_e$ is strictly diagonal, then $A_{e+1}$ is strictly diagonal?

---

> > > ### Author Response · Authors · 2023-11-22
> > >
> > > Thank you for your constructive comments! Your suggestion helps us to reconsider the mathematic principle behind our method. Specifically, we propose a rigorous theorem as follows,
> > >
> > > To avoid confusion in the proof between $\alpha$ and the elements in the matrix $A$, we temporarily denote the affine matrix $A$ as $N$.
> > >
> > > **Theorem 1.** When the stability factor $\alpha$ is small enough, if $N_{e}$ is strictly diagonally dominant, then $N_{e+1}$ is strictly diagonally dominant.
> > >
> > > **Proof:** Without loss of generality, we take the $i$-th row of $N_{e}$. Since $N_{e}$ is a strictly diagonally dominant matrix, we have,
> > > $$
> > > |n_{ii}^{e}|>\sum_{j\neq i}|n_{ij}^{e}|. \tag 6
> > > $$
> > >
> > > Where $n_{ii}^{e}$, $n_{ij}^{e}$ are the elements of the epoch $e$ in the $i$-th row, $i$-th column and $i$-th row, $j$-th column of the matrix $N$. According to the above Equation 3, the absolute value of the $i$-th diagonal element of the $e$+1 epoch of matrix $N$ is,
> > >
> > > $$
> > > \begin{align}
> > > |n_{ii}^{e+1}|&=|n_{ii}^{e}+\eta g_{ii}^{e} \frac{\partial L^{e}}{\partial n_{ii}^{e*}}|, \tag 7 \\\\
> > > &=|n_{ii}^{e}+\eta \frac{\partial L^{e}}{\partial n_{ii}^{e*}}|. \tag 8
> > > \end{align}
> > > $$
> > > Where $g_{ii}^{e}=1$, $n\_{ii}\^{e*}$ are the $i$-th diagonal elements of $GM$, $N\_{e}\^{\*}$ at epoch e, respectively. $L^{e}$ is the loss at epoch $e$. Further,
> > > $$
> > > |n_{ii}^{e+1}|=|n_{ii}^{0}+\eta \sum_{x=0}^e \frac{\partial L^{x}}{\partial n_{ii}^{x*}}|. \tag 9
> > > $$
> > > $n_{ii}^{0}$ is the scale when the matrix is initialized. Therefore, the diagonal values of the matrix $N$ are not equal to $0$ during the optimization process. Next we focus on the right-hand side of Equation 6 at epoch $e$+1. Similarly,
> > > $$
> > > \begin{align}
> > > \sum_{j\neq i} |n_{ij}^{e+1}|&=\sum_{j \neq i} |n_{ij}^{e} + \eta g_{ii}^{e}\frac{\partial L^{e}}{\partial n_{ij}^{e*}}|,   \tag {10} \\\\
> > > &=\sum_{j \neq i}|n_{ij}^{h}+\eta\sum_{x=h}^{e} g_{ii}^{x}\frac{\partial L^{x}}{\partial n_{ij}^{x*}}|.  \tag {11}
> > > \end{align}
> > > $$
> > > Where $1\leq h \leq e$ is the epoch at which $n_{ij}$ starts updating. In other words, $n_{ij}^{h}=0$, and as $h$ gets smaller $n_{ij}$ gets closer to the diagonal. In addition, $g_{ii}^{x}=\alpha$. Therefore, we have,
> > > $$
> > > \begin{align}
> > > \sum_{j\neq i} |n_{ij}^{e+1}|&= \eta\alpha\sum_{j \neq i}|\sum_{x=h}^{e} \frac{\partial L^{x}}{\partial n_{ij}^{x*}}|. \tag{12}
> > > \end{align}
> > > $$
> > > To make $\sum_{j\neq i} |n_{ij}^{e+1}| < |n_{ii}^{e+1}|$ , we let
> > > $$
> > > \begin{align}
> > > \eta\alpha\sum_{j \neq i}|\sum_{x=h}^{e} \frac{\partial L^{x}}{\partial n_{ij}^{x*}}|&< |n_{ii}^{0}+\eta \sum_{x=0}^e \frac{\partial L^{x}}{\partial n_{ii}^{x*}}|,  \tag{13} \\\\
> > > \alpha&<\frac{|n_{ii}^{0}+\eta \sum_{x=0}^e \frac{\partial L^{x}}{\partial n_{ii}^{x*}}|}{\eta\sum_{j \neq i}|\sum_{x=h}^{e} \frac{\partial L^{x}}{\partial n_{ij}^{x*}}|}.  \tag{14}
> > > \end{align}
> > > $$
> > > Thus, when the stability factor $\alpha$ is sufficiently small, if $N_{e}$ is a strictly diagonally dominant matrix, then $N_{e+1}$ is a strictly diagonally dominant matrix. The theorem is proved.

---

> ### Comment · Reviewer_gGYu · 2023-11-22
>
> Thank you for your response. I believe that incorporating an adaptive $\alpha$ could further enhance GM's performance. Specifically, I suggest considering bounding $\alpha$ at each iteration using the initial values of matrix $N$, the learning rate $\eta$, and the gradient value at epoch $h$. Given that $\eta$ decreases over epochs, adjusting the parameter $\alpha$ accordingly might provide additional benefits.

---

> ### Author Response · Authors · 2023-11-23
>
> Thank you for your suggestion. We will implement adaptive adjustments to the stability factor $\alpha$.

---

### Author Response · Authors · 2023-11-22

Most reviewers had questions about AffineQuant's training as well as inference overhead.

1. In terms of training costs, **Q**uantization **A**ware-**T**raining (QAT) for LLM involves the fine-tuning process to address quantization errors. LLM-QAT [1] requires 80 GPU hours to fine-tune the model for a single A100 on LLaMA-7B. The time taken to generate the data used for fine-tuning the model on LLaMA-7B is not included in this count. In comparison, AffineQuant quantizes LLaMA-7B in just 6.5 hours on a single A100. Similarly, for LLaMA-65B, AffineQuant takes only 78 hours to obtain the quantized model, which is comparable to LLM-QAT on LLaMA-7B. Despite the significantly expensive fine-tuning process, LLM-QAT does not demonstrate any advantage over AffineQuant in terms of experimental results. Please refer to the table below for further details.

|                     | Bits  | PIQA($\uparrow$) | ARC-e($\uparrow$) | WinoGrande($\uparrow$) | BoolQ($\uparrow$) | ARC-c($\uparrow$) | HellaSwag($\uparrow$) | Avg.($\uparrow$) |
| ------------------- | ----- | ---------------- | ----------------- | ---------------------- | ----------------- | ----------------- | --------------------- | ---------------- |
| FP16                | -     | 79.3             | 73.0              | 70.0                   | 76.8              | 48.0              | 76.1                  | 70.5             |
| LLM-QAT+SmoothQuant | 4-4-4 | 55.9             | 35.5              | 50.6                   | 62.4              | 26.4              | 47.8                  | 46.4             |
| AffineQuant         | 4-4-4 | 69.4             | 42.6              | 55.3                   | 63.7              | 31.9              | 57.7                  | 53.4             |

2. For the inference cost, we optimize the matrix after LayerNorm for diagonal elements only. This allows us to fuse the affine matrix with the LayerNorm weights and bias. For the remaining positions, we apply matrix transformations. After optimization, the matrix can be further fused with the weights and bias of the preceding linear layer. Consequently, AffineQuant can be achieved without introducing any additional overhead to model inference. We utilize the MLC-LLM [2] library to compile the LLaMA model at various scales under only-weight quantization configuration on a single A100, and the resulting inference speeds are presented in the following table.

| LLaMA     | 7B    | 7B    | 7B      | 13B   | 13B   | 13B     | 30B   | 30B   | 30B     | 65B   | 65B   | 65B     |
| --------- | ----- | ----- | ------- | ----- | ----- | ------- | ----- | ----- | ------- | ----- | ----- | ------- |
|           | WM    | RM    | token/s | WM    | RM    | token/s | WM    | RM    | token/s | WM    | RM    | token/s |
| FP        | 12.6G | 14.4G | 69.2    | 24.3G | 27.1G | 52.5    | 60.6G | 66.1G | 23.9    | OOM   | -     | -       |
| W4A16g128 | 3.8G  | 5.7G  | 155.3   | 7.0G  | 10.0G | 102.7   | 16.7G | 21.7G | 45.2    | 33.0G | 41.0G | 25.2    |
| W3A16g128 | 3.2G  | 5.1G  | 87.9    | 5.8G  | 8.7G  | 63.7    | 13.7G | 18.7G | 30.3    | 27.0G | 35.1G | 15.7    |
| W2A16g128 | 2.2G  | 4.1G  | 84.5    | 4.0G  | 7.5G  | 95.2    | 9.2G  | 14.1G | 38.3    | 18.0G | 25.6G | 25.6    |

---

> ### Author Response · Authors · 2023-11-22
>
> 3. Meanwhile, we compare the 4/4 bit quantization performance of LLaMA1&2 models on WikiText2 and C4 datasets in the following table. We evaluate the quantization performance of LLaMA-7B, 13B, 30B on six zero-shot datasets using 4/4 bit quantization in the following table. There was no additional inference overhead in either experiment.
>
> |                        | PIQA($\uparrow$) | ARC-e($\uparrow$) | WinoGrande($\uparrow$) | BoolQ($\uparrow$) | ARC-c($\uparrow$) | HellaSwag($\uparrow$) | Avg.($\uparrow$) |
> | ---------------------- | ---------------- | ----------------- | ---------------------- | ----------------- | ----------------- | --------------------- | ---------------- |
> | LLaMA-7B, OmniQuant    | 66.15            | 45.20             | 53.43                  | 63.51             | 31.14             | 56.44                 | 52.65            |
> | LLaMA-7B, AffineQuant  | 69.37            | 42.55             | 55.33                  | 63.73             | 31.91             | 57.65                 | 53.42            |
> | LLaMA-13B, OmniQuant   | 69.69            | 47.39             | 55.80                  | 62.84             | 33.10             | 58.96                 | 54.37            |
> | LLaMA-13B, AffineQuant | 66.32            | 43.90             | 54.70                  | 64.10             | 29.61             | 56.88                 | 52.58            |
> | LLaMA-30B, OmniQuant   | 71.21            | 49.45             | 59.19                  | 65.33             | 34.47             | 64.65                 | 56.63            |
> | LLaMA-30B, AffineQuant | 70.84            | 49.41             | 58.64                  | 70.12             | 37.12             | 65.53                 | 58.61            |
>
> |            | Methods     | WikiText2 | C4    |
> | ---------- | ----------- | --------- | ----- |
> | LLaMA-7B   | OmniQuant   | 11.26     | 14.51 |
> |            | AffineQuant | 10.28     | 13.64 |
> | LLaMA-13B  | OmniQuant   | 10.87     | 13.78 |
> |            | AffineQuant | 10.32     | 13.44 |
> | LLaMA-30B  | OmniQuant   | 10.33     | 12.49 |
> |            | AffineQuant | 9.35      | 11.58 |
> | LLaMA2-7B  | OmniQuant   | 14.26     | 18.02 |
> |            | AffineQuant | 12.69     | 15.76 |
> | LLaMA2-13B | OmniQuant   | 12.30     | 14.55 |
> |            | AffineQuant | 11.45     | 13.97 |
>
>
>
> [1] Liu et al., LLM-QAT: Data-Free Quantization Aware Training for Large Language Models.
>
> [2] MLC-LLM, https://github.com/mlc-ai/mlc-llm.

---

### Author Response · Authors · 2023-11-22

If there are any questions about the algorithm execution details, inference efficiency, GM implementation, affine matrix fusion process, please access this anonymous code link: https://github.com/anonymousiclr1842/AffineQuant. We fully open source the AffineQuant code.

---

### Meta-Review · Area_Chair_Bisb · 2023-12-12

**Metareview:**

Although not all reviewers were in favour of acceptance, most were, with a high score. This paper is more innovative than most papers in the crowded area of neural net compression, which tend to put together well-known ideas without any particular insight. The results appear to be good, and timely, in order to reduce the size of LLMs. Hence I recommend acceptance.

**Justification For Why Not Higher Score:**

N/A

**Justification For Why Not Lower Score:**

See metareview.

---

### Decision · Program_Chairs · 2024-01-16

Accept (poster)